# RePrompt: Reasoning-Augmented Reprompting for Text-to-Image Generation via Reinforcement Learning

**Mingrui Wu**[13][*]**Lu Wang**[2][†]**, Pu Zhao**[2][†]**, Fangkai Yang**[2][†]**, Jianjin Zhang**[2]**, Jianfeng Liu**[2]**,
Yuefeng Zhan**[2]**, Weihao Han**[2]**, Hao Sun**[2]**, Jiayi Ji**[1][‡]**, Xiaoshuai Sun**[1]**, Qingwei Lin**[2]**,
Weiwei Deng**[2]**, Dongmei Zhang**[2]**, Feng Sun**[2]**, Qi Zhang**[2]**, Rongrong Ji**[1]

1 Key Laboratory of Multimedia Trusted Perception and Efficient Computing,
Ministry of Education of China, Xiamen University, 361005, P.R. China.
2 Microsoft. 3 Zhongguancun Academy, Beijing, China.100094.

## Abstract

Despite recent progress in text-to-image (T2I) generation, existing models often struggle to faithfully capture user intentions from short and under-specified prompts. While prior work has attempted to enhance prompts using large language models (LLMs), these methods frequently generate stylistic or unrealistic content due to insufficient grounding in visual semantics and real-world composition. Inspired by recent advances in reasoning for language model, we propose RePrompt, a novel reprompting framework that introduces explicit reasoning into the prompt enhancement process via reinforcement learning. Instead of relying on handcrafted rules or stylistic rewrites, our method trains a language model to generate structured, self-reflective prompts by optimizing for image-level outcomes. The tailored reward models assesse the generated images in terms of human preference, semantic alignment, and visual composition, providing indirect supervision to refine prompt generation. Our approach enables end-to-end training without human-annotated data. Experiments on GenEval and T2I-Compbench show that RePrompt significantly boosts spatial layout fidelity and compositional generalization across diverse T2I backbones, establishing new state-of-the-art results. Code is available at: `https://github.com/microsoft/DKI_LLM/tree/main/RePrompt`.

## 1 Introduction

Text-to-image (T2I) generation has made rapid progress with the rise of large-scale generative models Labs (2024); Esser et al. (2024); Podell et al. (2023); Chen et al. (2024b), yet a persistent challenge remains: users typically provide concise and under-specified prompts, which often result in images that fail to reflect the intended semantics or visually coherent compositions. Generated outputs may misrepresent object counts, overlook spatial relations, or violate real-world plausibility. This misalignment arises from the gap between natural language descriptions and the structured visual logic required for faithful image generation Yang et al. (2024b).

Previous work on prompt enhancement in T2I can be divided into two main approaches. The first approach focuses on iterative refinement: an image is generated from an initial prompt, and subsequent feedback, derived either from human preference models or from automated scoring systems, is used to improve the prompt or intermediate representations over multiple rounds Yang et al. (2024d); Wu et al. (2024); Wang et al. (2024); Guo et al. (2025b). Although this approach can progressively improve image quality, it suffers from high latency and computational overhead due to repeated image generation, and it rarely incorporates explicit scene semantics or compositional reasoning. The second approach enriches prompts in a single pass by leveraging large language models (LLMs) to inject additional detail and context Betker et al. (2023); Hao et al. (2023). While these methods produce linguistically fluent and expressive descriptions, they frequently generate prompts that produce

---

[*]Work done at microsoft. [†]Project Leader. [‡]Corresponding author.

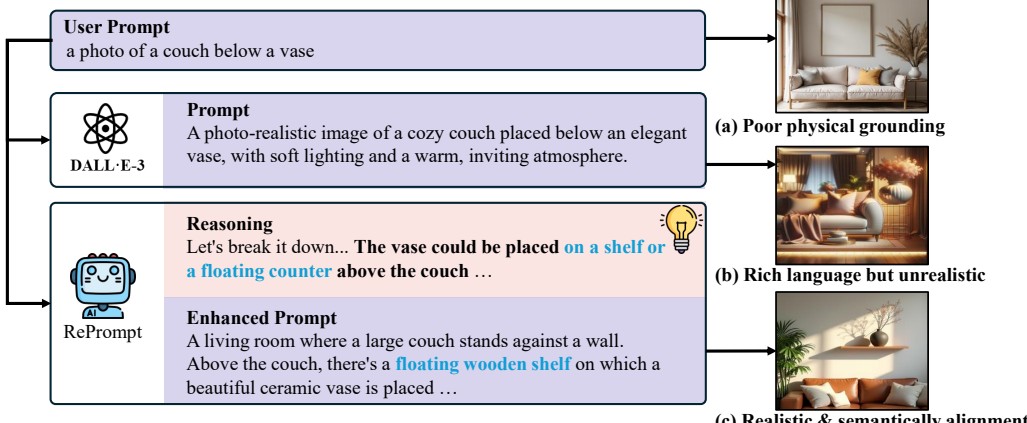

Figure 1: Given the user prompt "*a photo of a couch below a vase*", existing models like DELL-E3 generate rich language descriptions but often produce unrealistic or physically implausible compositions. In contrast, our RePrompt performs explicit chain-of-thought reasoning to resolve spatial relations, resulting in enhanced prompts that guide text-to-image models towards realistic and semantically aligned generations.

images with semantically inconsistent or visually implausible content, such as conflicting object placements or unrealistic interactions, because the underlying LLMs lack grounding in physical reality and do not incorporate feedback from downstream visual tasks. As a result, they frequently hallucinate content or miss critical spatial and attribute-level relationships (see Figure 1).

In contrast, we propose RePrompt, a reasoning-augmented prompt refinement framework trained via reinforcement learning. Rather than relying on stylistic rewriting or black-box completions, RePrompt trains a language model to generate structured and semantically grounded prompts through self-reflection and step-by-step decomposition. Motivated by recent advances in reasoning-augmented language models Guo et al. (2025a); Trung et al. (2024); Team et al. (2025); Jaech et al. (2024), RePrompt enables the model to internally simulate the visual implications of a prompt—much like how humans mentally visualize a scene before drawing. This structured, logic-driven process anticipates potential errors (e.g., conflicting object positions, missing entities, or spatial incoherence) during prompt construction, thereby reducing the need for multiple rounds of image generation.

A core component of RePrompt is a **T2I RePrompt Reward Model** tailored for text-to-image generation. Instead of relying on pre-labeled reasoning traces or handcrafted prompt templates, RePrompt learns from downstream visual feedback by optimizing prompt generation through reinforcement learning. To capture the multifaceted nature of image quality, we design a ensemble reward that evaluates generated images along three dimentions: human preference, visual realism, and semantic alignment with the input. By learning from diverse and grounded feedback signals, the model develops a more robust reasoning strategy that transfers across prompt types, scene structures, and T2I backbones, enabling stronger performance on unseen inputs without overfitting to specific linguistic patterns.

Experiments on GenEval Ghosh et al. (2023) and T2I-Compbench Huang et al. (2023) demonstrate that RePrompt significantly improves compositional accuracy, semantic fidelity, and spatial coherence. Notably, on the GenEval benchmark, RePrompt surpasses Qwen2.5 3B-enhanced baselines by +77.1% (FLUX Labs (2024)), +78.8% (SD3 Esser et al. (2024)) and +122.2% (Pixart-$\Sigma$ Chen et al. (2024b)) in the position category, highlighting its superior capability in grounding spatial relations. Furthermore, RePrompt achieves the best overall accuracy (0.76) among all evaluated methods while maintaining an order-of-magnitude lower latency than optimization-heavy baselines like Idea2Img (30s vs. 140s per image), offering a scalable and inference-efficient solution. These findings validate the effectiveness of explicit reasoning in prompt construction for closing the semantic-visual alignment gap in text-to-image generation, without relying on larger language models or expensive optimization at inference time.

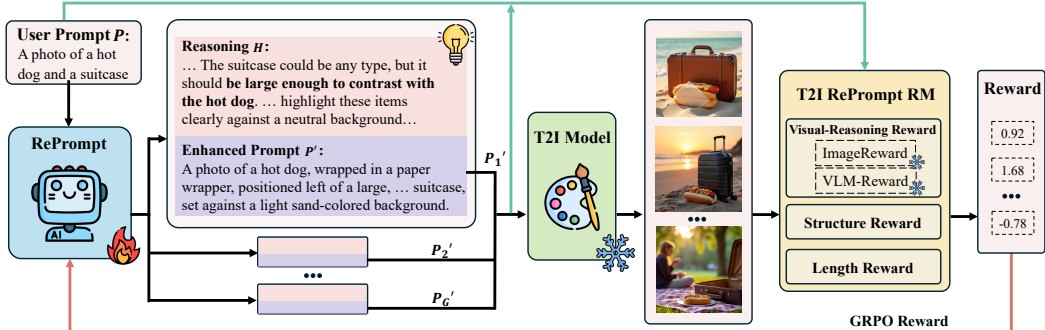

Figure 2: Overview of the proposed RePrompt. For each input prompt, RePrompt generates multiple reasoning trace and enhanced prompt pairs. The reasoning trace guides the model to produce more detailed, image-grounded prompts. These are used to synthesize candidate images via a T2I model, which are then scored by a reward model. Feedback is used to update RePrompt via GRPO.

## 2  RELATED WORK

**Text to Image Generation.** Recently, large-scale diffusion models Labs (2024); Esser et al. (2024); Podell et al. (2023); Chen et al. (2024b); Xie et al. (2025); Liu et al. (2024a); Betker et al. (2023); Ma et al. (2025a); Rombach et al. (2022); Saharia et al. (2022); Zhang et al. (2025c;b) have achieved impressive progress in generating high-resolution, photorealistic images from complex textual prompts. To enhance alignment between text and visuals, prior work has explored prompt engineering Hao et al. (2023); Mo et al. (2024); Yeh et al. (2024); Mañas et al. (2024); Yun et al. (2025); Cao et al. (2023); Qin et al. (2024); Yang et al. (2024d); Wu et al. (2024); Wang et al. (2024), often relying on manual or heuristic strategies with limited generalization. Some methods Liu et al. (2024b); He et al. (2024) utilize iterative feedback or black-box optimization to refine prompts over multiple steps. We propose a reinforcement learning-based framework that automatically refines prompts through iterative reasoning, achieving better semantic alignment than static or rule-based methods.

**Reinforcement Learning.** Reinforcement learning (RL) has proven effective in scenarios where iterative feedback is essential for task optimization. In the realm of generative models, RL-based approaches Guo et al. (2025a); Wallace et al. (2024); Yang et al. (2024c); Guo et al. (2024); Gupta et al. (2025); Lee et al. (2024); Zhao et al. (2024); Nabati et al. (2024); Black et al. (2023); Liang et al. (2024); Kirstain et al. (2023); Lee et al. (2023); Shen et al. (2025); Zhang et al. (2024); Wang et al. (2025a); Jiang et al. (2025); Xue et al. (2025) have been employed to fine-tune outputs by maximizing a reward function that encapsulates desired attributes such as image realism, diversity, and semantic fidelity. In our framework, we adopt RL techniques to drive the automatic optimization of text prompts. By defining a multi-faceted reward that not only evaluates the visual quality of the generated image but also the interpretability and relevance of the prompt, our approach enables the model to learn an optimal prompt refinement strategy over successive iterations. Notably, T2I-R1 Jiang et al. (2025) is closely related to our work, but it targets Janus-Pro Chen et al. (2025), a unified vision-language model. In contrast, our method trains an auxiliary LLM that generalizes across diverse text-to-image models, offering a more flexible Ma et al. (2023; 2024) and model-agnostic solution.

**Reasoning in LLM.** Reasoning in LLMs Wei et al. (2022) improves complex task solving by decomposing problems into intermediate steps Feng et al. (2025); Huang et al. (2025b); Yang et al. (2025); Zhang et al. (2025a); Huang et al. (2025a); Yu et al. (2025); Ma et al. (2025b); Li et al. (2025); Lu et al. (2025). In multimodal generation, reasoning mechanisms Yang et al. (2024b); Guo et al. (2025b); Wang et al. (2025c); Zhang et al. (2025d); Sahili et al. (2024); Chen et al. (2024a) enhance prompt understanding and semantic alignment. Our method integrates reasoning with reinforcement learning, enabling step-wise prompt refinement that improves text-image alignment, and image quality—offering a new perspective for prompt optimization in T2I generation.

## 3 METHOD

We present **RePrompt**, a reasoning-augmented reprompting framework for text-to-image (T2I) generation. RePrompt decouples prompt generation from image generation, i.e., training a language model to produce structured, semantically rich prompts, while keeping the T2I backbone fixed. We optimize RePrompt via reinforcement learning (RL) to directly improve downstream image quality, compositional correctness, and usability.

### 3.1 FRAMEWORK OVERVIEW

RePrompt comprises three main modules (Figure 2): 1) a **Prompting Policy** $\pi_\theta$, which produces a *reasoning trace H* and an enhanced prompt $P'$; 2) a fixed **T2I Synthesizer** $f_\phi$, which renders an image $I$ from $P'$; 3) a **T2I RePrompt Reward Model** $R_{\text{total}}(I, P, P')$, which scores the image on realism, semantic alignment, and prompt structure.

Given an input prompt $P$, the policy samples: $y = (H, P') \sim \pi_\theta(P)$, the synthesizer then generates: $I = f_\phi(P')$. Since $f_\phi$ is non-differentiable, we formulate prompt generation as a single-step Markov Decision Process (MDP): **State**: the original prompt $P$. **Action**: sampling $y = (H, P') \sim \pi_\theta(y \mid P)$. **Transition**: deterministic mapping $P' \mapsto I = f_\phi(P')$. **Reward** $r$: the reasoning reward $r = R_{\text{total}}(I, P, P')$. **Objective**: maximize $E_{P \sim \mathcal{D}}\big[E_{y \sim \pi_\theta(y|P)}[r]\big]$.

We train $\pi_\theta$ via reinforcement learning (Group Relative Policy Optimization, GRPO Guo et al. (2025a)) to improve downstream image quality. By keeping the T2I model $f_\phi$ fixed, RePrompt learns backbone-specific reasoning strategies that enhance semantic fidelity and visual realism without requiring any manually annotated reasoning traces.

### 3.2 T2I REPROMPT REWARD MODEL

A central component of our framework is the **T2I RePrompt Reward Model**—an ensemble, image-grounded reward function specifically designed for the prompt refinement task in T2I generation. In contrast to generic reward functions used in natural language or vision tasks, our reward model is co-developed with the objective of enhancing reasoning-driven prompt construction. It explicitly evaluates whether a generated prompt yields an image that is *realistic*, *semantically faithful to the user intent*, and *compositionally coherent*.

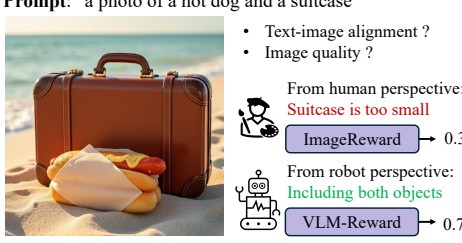

Figure 3: The Visual-Reasoning Reward.

This reward framework is critical to training RePrompt effectively and serves three key goals: (1) **Stable optimization:** Each component provides dense and structured feedback, mitigating the challenges of sparse or noisy reward signals during early-stage learning. (2) **Multi-faceted supervision:** The reward captures complementary aspects of T2I quality, including human preference, visual quality, and semantic alignment, ensuring holistic prompt improvement. (3) **Cross-model generalization:** Because the reward depends only on the prompt-image output pair and not on any specific T2I architecture, it generalizes naturally across different generation backbones and unseen prompt distributions. Together, these properties enable our reward model to not only guide the training of the reprompting policy but also ensure broad applicability and stable learning across varying settings in text-to-image generation.

**Visual-Reasoning Reward ($R_{\text{vis}}$).** This component (as shown in Figure 3) captures both user-aligned preferences and semantic correctness at the image level:

$$R_{\text{vis}} = \alpha\, R_{\text{pref}}^{\text{IMG}} + \gamma\, R_{\text{sem}}^{\text{VLM}}. \tag{1}$$

Here, $R_{\text{pref}}^{\text{IMG}}$ is derived from ImageReward Xu et al. (2023), which used to evaluate whether generated images align with human preferences. $R_{\text{sem}}^{\text{VLM}}$ is obtained from VLM-Reward Achiam et al. (2023), which evaluates semantic consistency and visual quality using a vision-language model. The weights $\alpha$ and $\gamma$ allow us to control the trade-off between perceived image quality and factual alignment.

**Structure Reward ($R_{\text{struc}}$).**    To ensure that the generated output maintains a clear reasoning-to-prompt format, we enforce a structured syntax:

<reason>...</reason><prompt>...</prompt>

We apply a binary reward:

$$R_{\text{struc}} = \begin{cases} +1, & \text{if format is correct} \\ -1, & \text{otherwise} \end{cases} \tag{2}$$

This reward encourages models to adhere to a standardized output layout, simplifying downstream parsing and ensuring the interpretability of reasoning traces.

**Length Reward ($R_{\text{len}}$).**    To ensure compatibility with real-world T2I models such as SDXL and FLUX.1, which impose token-length limits, we apply a constraint on prompt length:

$$R_{\text{len}} = \begin{cases} +1, & L_{\min} \le L \le L_{\max}, \\ -1, & \text{otherwise} \end{cases} \tag{3}$$

where $L$ is the token length of $P'$, and $[L_{\min}, L_{\max}]$ is empirically set to 15 and 77 tokens. This ensures the prompt is both concise and informative.

**Ensemble Reward and Optimization.**    All reward components are normalized to unit variance and summed:

$$R_{\text{total}} = R_{\text{vis}} + R_{\text{struc}} + R_{\text{len}}. \tag{4}$$

The RePrompt policy $\pi_\theta$ is trained to maximize expected downstream reward:

$$\theta^* = \arg\max_\theta \; \mathbb{E}_{P \sim \mathcal{D}, \, y \sim \pi_\theta(y|P)} \left[ R_{\text{total}}(I, P, P') \right], \tag{5}$$

where $I = f_\phi(P')$ and $f_\phi$ is a fixed text-to-image generator.

**Generalization and Flexibility.**    Since the reward depends only on the input-output behavior of the system, not the internals of the image model, our framework generalizes well across unseen prompts and new T2I backbones. This modularity also enables RePrompt to adapt to model-specific strengths and failure patterns, while maintaining stable and interpretable training signals throughout.

### 3.3    REPROMPT OPTIMIZATION

Prompt generation for T2I involves a non-differentiable, black-box image renderer $f_\phi$. RL allows us to optimize $\pi_\theta$ directly with respect to *downstream image outcomes* rather than proxy losses on text. Moreover, because RePrompt is *individualized to each T2I backbone*, it can adapt its reasoning and prompt style to the specific strengths and limitations of a given model—improving generalization and image fidelity without retraining $f_\phi$.

At each update step, for a given user prompt $P$, we sample a set of $G$ candidate outputs $\{y_i\}_{i=1}^{G} \sim \pi_{\theta_{\text{old}}}(y \mid P)$. Each candidate $y_i = (H_i, P_i')$ yields an image $I_i = f_\phi(P_i')$ and receives a scalar reward $r_i = R_{\text{total}}(I_i, P, P')$. We then compute normalized advantages:

$$A_i = \frac{r_i - \mu_r}{\sigma_r}, \quad \mu_r = \frac{1}{G} \sum_{j=1}^{G} r_j, \quad \sigma_r = \sqrt{\frac{1}{G} \sum_{j=1}^{G} (r_j - \mu_r)^2}. \tag{6}$$

We use Group Relative Policy Optimization (GRPO) Guo et al. (2025a) as a practical and stable update method for training the reprompting policy based on group-wise reward comparisons. The objective is defined as:

$$\mathcal{J}_{\text{GRPO}}(\theta) = E_{P, \{y_i\} \sim \pi_{\theta_{\text{old}}}} \left[ \frac{1}{G} \sum_{i=1}^{G} \min\left( \rho_i A_i, \; \text{clip}(\rho_i, 1 - \varepsilon, 1 + \varepsilon) A_i \right) \right.$$
$$\left. - \beta_{\text{KL}} \, KL\big( \pi_\theta(y \mid P) \, \| \, \pi_{\text{ref}}(y \mid P) \big) \right], \tag{7}$$

where $\rho_i = \frac{\pi_\theta(y_i|P)}{\pi_{\theta_{\mathrm{old}}}(y_i|P)}$, $\varepsilon$ and $\beta_{\mathrm{KL}}$ are clipping and penalty coefficients, $\pi_{\mathrm{ref}}$ is a reference policy (e.g., the initial or distillation policy). By fixing $f_\phi$ and optimizing only $\pi_\theta$, RePrompt can be applied *universally* to any off-the-shelf T2I model, learning backbone, specific reasoning and prompt strategies without retraining the image generator. We further validate RePrompt with a variance-reduction analysis, showing that structured reasoning reduces reward uncertainty and lowers the sample complexity for accurate estimation. This leads to faster and more stable GRPO training. Full analysis and proof are provided in Appendix B.

## 4 EXPERIMENTS

### 4.1 SETTINGS

**Implementation Details.** We use Qwen2.5-3B Yang et al. (2024a) as our base language model. For the text-to-image model used for training, we use the FLUX.1-dev Labs (2024) model, which generates images at a resolution of 512×512 pixels. Our model is trained using the TRL [1] reinforcement learning framework for 3 epochs, with 4 outputs generated per instance, the weight of ImageReward Xu et al. (2023) and VLM-Reward are both 0.5. The VLM used for computing VLM-Reward is GPT-4V. All experiments were conducted on 8 NVIDIA A100 (80GB) GPUs, and the entire training process required about 6 hours. More details are in the Appendix C.

**Training Data.** Inspired by the prompt construction strategy in GenEval Ghosh et al. (2023), we adapt six object-centric templates to a newly curated list of 288 common daily objects generated via GPT-4 Achiam et al. (2023). This results in a training corpus of 9,000 prompts, carefully filtered to avoid overlap with the GenEval. We use 8,000 prompts to fine-tune our RePrompt via supervised learning, and 1,000 prompts for reinforcement learning.

**Evaluation Setup.** To assess RePrompt, we evaluate on two benchmarks: GenEval Ghosh et al. (2023) and T2I-Compbench Huang et al. (2023). GenEval focuses on instance-level alignment with user intent using concise prompts, while T2I-Compbench measures compositional generation under complex scenarios involving multiple objects, attributes, and spatial relations.

### 4.2 COMPARISION

Table 1 presents the performance comparison across various text-to-image generation models evaluated on the GenEval benchmark, which assesses six fine-grained composition capabilities: single-object, two-object, counting, color, spatial position, and attribute binding.

Notably, our method demonstrates exceptional gains in spatial layout understanding (Position). For instance, when built upon FLUX Labs (2024), our approach achieves a 0.62 score on Position, representing a +77.1% relative improvement over the Qwen2.5 3B baseline. Similarly, for SD3 Esser et al. (2024), we observe a +78.8% gain, and for Pixart-$\Sigma$ Chen et al. (2024b), the relative improvement reaches an impressive +122.2%. This strong enhancement highlights the strength of our compositional training strategy in explicitly grounding spatial relations between objects. Beyond the Position metric, our method also achieves substantial improvements in Counting (+22.2% for FLUX, +13.2% for SD3, +16.7% for Pixart-$\Sigma$) and moderate gains in attribute binding (+2.1% to +9.4%). As a result, we observe consistent boosts in overall GenEval scores: +11.8% (FLUX), +10.3% (SD3), and +6.9% (Pixart-$\Sigma$). These results verify the effectiveness of our reinforcement learning–based reprompting strategy, which enables the language model to iteratively reason about visual composition and generate more precise, image-aligned prompts, and brings state-of-the-art advances in spatial understanding.

### 4.3 GENERALIZATION PERFORMANCE ON T2I-COMPBENCH

We further assess the robustness of our method on the T2I-Compbench benchmark, which tests compositional generalization across six dimensions: color, shape, texture, spatial reasoning, numeracy, and complex attribute combinations. As shown in Table 2, RePrompt consistently improves performance across all evaluated backbones. In particular, it significantly boosts spatial

---

[1]https://github.com/huggingface/trl

Table 1: Evaluation of text-to-image generation on the GenEval benchmark. Our method consistently outperforms strong baselines, achieving the best overall scores. Notably, our approach shows substantial gains in spatial position understanding over Qwen2.5 3B-enhanced baselines, demonstrating its superior capability in grounding spatial relations.

| Method | Single object | Two object | Counting | Colors | Position | Attribute binding | Overall ↑ |
|---|---|---|---|---|---|---|---|
| FLUX Labs (2024) | 0.99 | 0.79 | 0.75 | 0.78 | 0.18 | 0.45 | 0.66 |
| + Promptist Hao et al. (2023) | 0.98 | 0.72 | 0.70 | 0.78 | 0.21 | 0.44 | 0.66 |
| + PAG Yun et al. (2025) | 0.97 | 0.74 | 0.73 | 0.80 | 0.36 | 0.46 | 0.69 |
| + GPT4 | 0.99 | 0.79 | 0.68 | 0.84 | 0.51 | 0.52 | 0.72 |
| + GPT5 | 1.00 | 0.81 | 0.70 | 0.85 | 0.51 | 0.51 | 0.73 |
| + GPTo3 | 0.98 | 0.76 | 0.65 | 0.83 | 0.48 | 0.50 | 0.70 |
| + Deepseek-r1 | 1.00 | 0.81 | 0.56 | 0.78 | 0.47 | 0.43 | 0.67 |
| + Qwen2.5 7B | 0.99 | 0.83 | 0.62 | 0.84 | 0.36 | 0.51 | 0.69 |
| + Qwen2.5 3B | 0.99 | 0.84 | 0.63 | 0.81 | 0.35 | 0.48 | 0.68 |
| + Ours (train w/ FLUX) | 0.98 | **0.87** | **0.77** | **0.85** | **0.62** | **0.49** | **0.76** |
| *Improvement over Qwen2.5 3B* | *-1.0%* | *+3.6%* | *+22.2%* | *+4.9%* | ***+77.1%*** | *+2.1%* | *+11.8%* |
| SD3 Esser et al. (2024) | 1.00 | 0.85 | 0.62 | 0.88 | 0.22 | 0.58 | 0.69 |
| + Promptist | 0.99 | 0.84 | 0.66 | 0.84 | 0.45 | 0.52 | 0.69 |
| + PAG | 0.99 | 0.85 | 0.68 | 0.85 | 0.49 | 0.53 | 0.71 |
| + GPT4 | 1.00 | 0.84 | 0.51 | 0.85 | 0.48 | 0.54 | 0.70 |
| + GPT5 | 1.00 | 0.85 | 0.53 | 0.85 | 0.47 | 0.50 | 0.70 |
| + GPTo3 | 0.99 | 0.82 | 0.50 | 0.84 | 0.41 | 0.52 | 0.68 |
| + Deepseek-r1 | 0.99 | 0.82 | 0.53 | 0.80 | 0.44 | 0.46 | 0.67 |
| + Qwen2.5 7B | 1.00 | 0.82 | 0.49 | 0.85 | 0.34 | 0.58 | 0.68 |
| + Qwen2.5 3B | 1.00 | 0.86 | 0.53 | 0.84 | 0.33 | 0.55 | 0.68 |
| + Ours (train w/ FLUX) | 0.99 | **0.86** | 0.60 | 0.86 | **0.59** | **0.60** | **0.75** |
| *Improvement over Qwen2.5 3B* | *-1.0%* | *0.0%* | *+13.2%* | *+2.4%* | ***+78.8%*** | *+9.1%* | *+10.3%* |
| Pixart-Σ Chen et al. (2024b) | 0.99 | 0.60 | 0.47 | 0.81 | 0.10 | 0.26 | 0.54 |
| + Promptist | 0.98 | 0.60 | 0.49 | 0.80 | 0.20 | 0.27 | 0.55 |
| + PAG | 0.98 | 0.63 | 0.52 | 0.80 | 0.28 | 0.29 | 0.56 |
| + GPT4 | 0.96 | 0.67 | 0.48 | 0.84 | 0.36 | 0.31 | 0.60 |
| + GPT5 | 0.97 | 0.68 | 0.49 | 0.84 | 0.37 | 0.31 | 0.61 |
| + GPTo3 | 0.96 | 0.67 | 0.48 | 0.83 | 0.35 | 0.31 | 0.60 |
| + Deepseek-r1 | 0.99 | 0.63 | 0.43 | 0.78 | 0.24 | 0.27 | 0.56 |
| + Qwen2.5 7B | 0.96 | 0.67 | 0.43 | 0.83 | 0.20 | 0.32 | 0.57 |
| + Qwen2.5 3B | 0.99 | 0.68 | 0.48 | 0.82 | 0.18 | 0.32 | 0.58 |
| + Ours (train w/ FLUX) | 0.98 | 0.64 | **0.56** | **0.81** | **0.40** | **0.35** | **0.62** |
| *Improvement over Qwen2.5 3B* | *-1.0%* | *-5.9%* | *+16.7%* | *-1.2%* | ***+122.2%*** | *+9.4%* | *+6.9%* |

Table 2: Evaluation of text-to-image generation on the T2I-Compbench. We report the baseline results, their variants enhanced with Qwen2.5 3B, and our method trained with FLUX. Our approach consistently improves performance across most aspects, particularly in Spatial compositions.

| Method | Color | Shape | Texture | Spatial | Numeracy | Complex |
|---|---|---|---|---|---|---|
| FLUX | 0.7223 | 0.4796 | 0.6522 | 0.2494 | 0.6101 | 0.3616 |
| + Qwen2.5 3B | 0.7149 | 0.5103 | 0.6012 | 0.2579 | 0.5982 | 0.3325 |
| + Ours (train w/ FLUX) | **0.7501** | **0.5276** | 0.6515 | **0.3301** | **0.6499** | **0.3721** |
| SD3 | 0.7941 | 0.5812 | 0.7224 | 0.2815 | 0.5871 | 0.3714 |
| + Qwen2.5 3B | 0.7227 | 0.5478 | 0.6581 | 0.2549 | 0.5934 | 0.3307 |
| + Ours (train w/ FLUX) | 0.7866 | **0.5891** | 0.7184 | **0.3315** | **0.6289** | **0.3744** |
| Pixart-Σ | 0.5682 | 0.4717 | 0.5622 | 0.2497 | 0.5366 | 0.3655 |
| + Qwen2.5 3B | 0.6618 | 0.4814 | 0.5662 | 0.2481 | 0.5443 | 0.3335 |
| + Ours (train w/ FLUX) | **0.6665** | **0.5011** | **0.6190** | **0.2913** | **0.5716** | **0.3680** |

compositional scores (e.g., from 0.2494 to 0.3301 on FLUX and from 0.2815 to 0.3315 on SD3) and enhances numeracy understanding, two long-standing challenges in text-to-image generation. Moreover, our method outperforms stronger LLM-enhanced baselines, for instance, on Pixart-Σ,

Table 3: Ablation study of SFT and RL on the GenEval benchmark.

| Method | Single object | Two object | Counting | Colors | Position | Attribute binding | Overall ↑ |
|---|---|---|---|---|---|---|---|
| FLUX Labs (2024) | 0.99 | 0.79 | 0.75 | 0.78 | 0.18 | 0.45 | 0.66 |
| + Qwen2.5 3B | 0.99 | 0.84 | 0.63 | 0.81 | 0.35 | 0.48 | 0.68 |
| w/ SFT | 0.99 | 0.83 | 0.64 | 0.81 | 0.43 | 0.44 | 0.69 |
| w/ RL | 0.98 | 0.83 | 0.71 | **0.87** | 0.41 | **0.53** | 0.72 |
| w/ SFT + RL | 0.98 | **0.87** | **0.77** | 0.85 | **0.62** | 0.49 | **0.76** |

Table 4: Quantitative comparison between RePrompt and other method on image generation accuracy and latency with the subset of Geneval. All latency is measured on a single NVIDIA A100 GPU.

| Method | Accuracy ↑ | Latency (s / img) ↓ |
|---|---|---|
| FLUX | 0.65 | 20 |
| Show-o Xie et al. (2024) | 0.55 | 3 |
| Idea2Img Yang et al. (2024d) (w/ FLUX) | 0.69 | 140 |
| PARM++ Guo et al. (2025b) (w/ Show-o) | 0.72 | 110 |
| **RePrompt** (w/ FLUX) | **0.76** | **30** |

RePrompt achieves notable gains in both texture and spatial dimensions. These results demonstrate that our approach generalizes well across diverse models and compositional skills, validating its effectiveness as a versatile and plug-and-play enhancement for real-world T2I systems.

## 4.4 ABLATION STUDY

**Ablation on SFT and RL.** Table 3 shows the effect of supervised fine-tuning (SFT) and reinforcement learning (RL) on GenEval. SFT brings modest gains (+0.01 overall), mainly improving positional understanding (0.35→0.43), suggesting it helps inject object-attribute knowledge but struggles with complex reasoning. RL yields larger boosts (+0.04 overall), as it directly optimizes visual correctness. Combining SFT and RL achieves the best results (0.76 overall), with strong improvements in spatial reasoning (0.62) and counting (0.77). These results confirm that SFT offers useful priors, while RL is key for compositional robustness.

**Ablation Study on Reasoning.** Table 5 presents an ablation study on the GenEval benchmark to assess the impact of reasoning in our reinforcement learning framework. The FLUX baseline, integrating a large language model (+Qwen2.5 3B) brings modest gains across most categories, raising the overall score to 0.68. Applying RL without reasoning achieves a similar overall improvement (0.68), suggesting that reward-driven optimization alone contributes to better alignment. However, incorporating reasoning into the RL loop leads to a more substantial improvement, pushing the overall score to 0.72. Notably, categories that require more complex semantic understanding—such as "Colors" (from 0.83 to 0.87) and "Attribute binding" (from 0.46 to 0.53)—benefit the most. These results demonstrate that step-by-step reasoning helps the model better decompose and interpret textual prompts, thereby enabling more accurate and faithful image generation.

**Comparison on Accuracy and Latency.** Table 4 benchmarks our RePrompt against prior methods in terms of both generation accuracy (GenEval overall) and inference latency. Among base models, FLUX.1 achieves higher accuracy compared to Show-o but suffers from a longer latency (20s vs. 3s per image), reflecting a trade-off between quality and speed. When incorporating advanced prompting techniques, Idea2Img (w/ FLUX.1) improves accuracy to 0.69 but at the cost of a significant latency increase (140s per image), while PARM++ (w/ Show-o) achieves 0.72 accuracy with 110s latency. In contrast, RePrompt achieves the best accuracy while maintaining a much lower latency. This demonstrates the effectiveness of RePrompt in enabling precise visual grounding without compromising efficiency, making it more practical for real-world deployment.

**Ablation Study on Trained T2I Models.** Figure 4 illustrates the generalization ability of RePrompt across three diverse T2I backbones: FLUX, SD3, and PixArt-Σ. Our method consistently boosts compositional scores on GenEval—improving FLUX from 0.66 to 0.76, SD3 from 0.69 to 0.75, and PixArt-Σ from 0.54 to 0.62—outperforming both baselines and the +Qwen2.5 variants. These

Table 5: Ablation study of reasoning on the GenEval benchmark.

| Method | Single object | Two object | Counting | Colors | Position | Attribute binding | Overall ↑ |
|---|---|---|---|---|---|---|---|
| FLUX | 0.99 | 0.79 | 0.75 | 0.78 | 0.18 | 0.45 | 0.66 |
| +Qwen2.5 3B | 0.99 | 0.84 | 0.63 | 0.81 | 0.35 | 0.48 | 0.68 |
| RL w/o reasoning | 1.00 | 0.81 | 0.68 | 0.83 | 0.33 | 0.46 | 0.68 |
| RL w/ reasoning | 0.98 | 0.83 | 0.71 | **0.87** | 0.41 | **0.53** | 0.72 |

Figure 4: Impact of our method across different base T2I models on the GenEval benchmark. Our method consistently improves the compositional understanding across all base models.

results demonstrate RePrompt's robustness and model-agnostic design, working effectively across both large (e.g., SD3) and lightweight (e.g., PixArt-Σ) backbones. Notably, we observe that stronger T2I models used during training lead to better generalization at inference, likely due to richer reasoning patterns induced during learning.

**Qualitative Comparison.** Figure 5 illustrates the qualitative advantage of RePrompt over existing T2I models. Baseline models often generate images with incorrect spatial relations or hallucinated objects. For example, when prompted with "a fire hydrant with a tennis racket," DALL-E 3 produces unrealistic, stylistic blends where the objects are merged. In contrast, RePrompt accurately grounds the tennis racket "around the base" of the fire hydrant, respecting the intended composition. Similarly, for "a photo of a dog above a cow," our method correctly depicts the dog in the air above the cow with "a high angle shot", aligning with the prompt semantics. These results highlight RePrompt's ability to mitigate typical failures in spatial arrangement and object interaction by generating prompts with explicit compositional cues. More cases are in the Appendix F.

**More Ablation Study.** We present additional ablation studies on training dynamics, and reward functions in the Appendix D to further validate the effectiveness of our reasoning design and reward selection, as well as the stability of the training process.

## 5 CONCLUSION

In this work, we propose a simple yet effective method to enhance compositional understanding in text-to-image (T2I) generation models. By injecting chain-of-thought (CoT) reasoning into the prompt construction pipeline and pairing each CoT with an enhanced prompt, our approach improves the alignment between textual descriptions and generated images. Extensive experiments on the GenEval benchmark demonstrate that our method consistently improves performance across various T2I backbones, including FLUX, SD3, and Pixart-Σ, without requiring additional model retraining. These results highlight the generalizability and plug-and-play nature of our method. We believe our approach offers a new perspective for improving controllability and compositional fidelity in generative models.

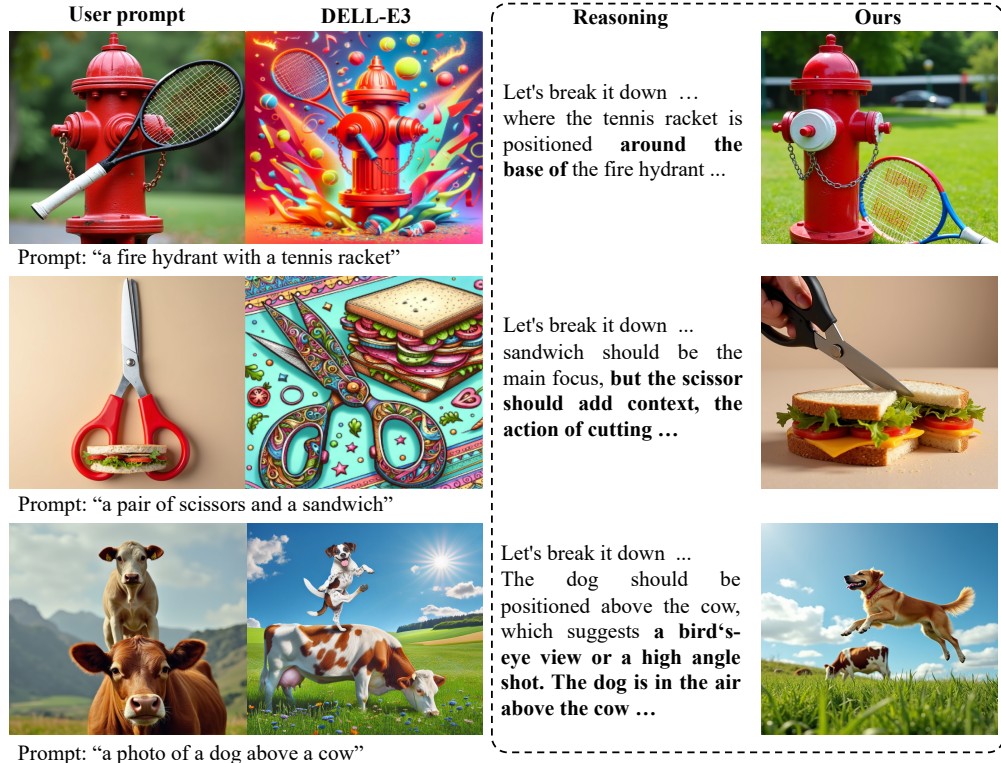

Figure 5: Qualitative results on compositional prompts. Compared to vanilla T2I models, our Re-Prompt improves spatial layout and object relations by generating enhanced prompts with explicit reasoning, leading to more faithful compositions.

ACKNOWLEDGMENTS

This work was supported by the National Key Research and Development Program of China (No. 2025YFE0113500), the National Science Fund for Distinguished Young Scholars (No.62025603), the National Natural Science Foundation of China (No. U22B2051, No. 62302411), and the Zhongguancun Academy, Beijing, China (No. 20240103).

## 6   ETHICS STATEMENT

We adhere to the ICLR Code of Ethics. This work does not involve human subjects, sensitive data, or harmful applications. Dataset usage complies with all relevant licenses. The authors declare no conflicts of interest. We are committed to fairness, privacy, and security, and confirm that all ethical standards and institutional requirements have been met.

## 7   REPRODUCIBILITY STATEMENT

We have taken several steps to ensure the reproducibility of our results. Detailed descriptions of the model architecture, training procedures, and experimental settings are provided in the main text and supplementary materials. All assumptions and derivations underlying our theoretical claims are clearly explained and fully proven in the appendix. The datasets used in our experiments are publicly available. Additionally, we provide anonymized source code as supplementary material to facilitate replication of our experiments. We encourage readers to refer to these materials for a complete understanding of our methodology and experimental setup.

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

APPENDIX

## A    BROADER IMPACT

Our work aims to enhance the controllability and fidelity of text-to-image generation by aligning prompt engineering with user intent through self-reflective reasoning. This could have a wide range of beneficial applications, such as improving accessibility tools for the visually impaired, assisting designers in rapid prototyping, or helping educators generate custom visual teaching materials. By reducing the gap between user intentions and generated images, our method empowers non-expert users to communicate complex visual ideas more effectively.

However, like many generative technologies, our system also presents potential risks. Enhanced prompts with richer semantics could inadvertently be exploited to generate more realistic harmful or misleading content, especially in the context of misinformation or deepfakes. To mitigate such risks, we recommend pairing our method with content moderation filters and safety alignment mechanisms in deployment. Additionally, since our method involves reinforcement learning with learned reward models, any bias in the reward signal may propagate into the prompt generation. Future research should explore fairness-aware training objectives and human-in-the-loop evaluation to ensure responsible use.

Overall, we believe that RePrompt contributes to the development of more interpretable, controllable, and user-aligned generative models, which are essential for trustworthy AI applications.

## B    VARIANCE REDUCTION VIA STRUCTURED REASONING

In this appendix, we provide a full theoretical analysis showing that conditioning prompt generation on an explicit reasoning trace $H$ strictly reduces the variance of the downstream reward estimator. This reduction in variance directly leads to improved sample efficiency for reinforcement learning.

### B.1    SETUP AND NOTATION

Let:

- $\mathcal{P}$ be the space of bare prompts $P'$.
- $\mathcal{H}$ be the space of reasoning traces $H$.
- $r : \mathcal{P} \to [0, R_{\max}]$ be the reward random variable obtained by sampling $P' \sim \pi(P)$ and generating an image $I = f_\phi(P')$.
- $r_H : \mathcal{H} \times \mathcal{P} \to [0, R_{\max}]$ be the reward when first sampling $H \sim \pi_H(P)$, then $P' \sim \pi(P \mid H)$, and finally $I = f_\phi(P')$.
- All expectations and variances are taken over the joint sampling of $H$ and $P'$.

### B.2    LAW OF TOTAL VARIANCE

By the law of total variance,

$$\mathrm{Var}\big[r(P')\big] = E_H\big[\mathrm{Var}[r(P') \mid H]\big] + \mathrm{Var}_H\big[E\big[r(P') \mid H\big]\big]$$

Since variances are nonnegative, we immediately have:

**Theorem B.1** (Variance Reduction)**.**

$$\mathrm{Var}\big[r(H, P')\big] = E_H\big[\mathrm{Var}\big[r \mid H\big]\big] \ \leq \ \mathrm{Var}\big[r(P')\big].$$

*Proof.* By definition,

$$\mathrm{Var}\big[r(H, P')\big] = E_H[\mathrm{Var}[r \mid H]]$$

and since

$$\mathrm{Var}[r(P')] = E_H[\mathrm{Var}[r \mid H]] + \mathrm{Var}_H[E[r \mid H]],$$

the nonnegativity of $\mathrm{Var}_H[E[r \mid H]]$ yields the result. $\square$

### B.3 Sample Complexity Improvement

Variance directly impacts the number of samples required to estimate the expected reward within a given accuracy. Consider estimating the expected reward $\mu = E[r]$ by drawing $N$ independent samples $\{r_i\}$. By Chebyshev's inequality,

$$\Pr\left(|\hat{\mu} - \mu| \geq \varepsilon\right) \ \leq \ \frac{\mathrm{Var}[r]}{N\varepsilon^2}.$$

Thus, to guarantee $\Pr(|\hat{\mu} - \mu| \geq \varepsilon) \leq \delta$, we require

$$N \ \geq \ \frac{\mathrm{Var}[r]}{\varepsilon^2\,\delta}.$$

Applying Theorem B.1, using reasoning reduces the required sample size:

$$N_{\text{reasoning}} \ = \ \frac{\mathrm{Var}[r(H, P')]}{\varepsilon^2\delta} \ \leq \ \frac{\mathrm{Var}[r(P')]}{\varepsilon^2\delta} \ = \ N_{\text{bare}}.$$

### B.4 Discussion

This analysis formalizes the intuition that introducing an intermediate reasoning trace $H$ conditions the policy on structured, compositional information about the scene, thereby reducing uncertainty (variance) in the reward signal. Empirically, this translates to fewer rollouts needed during GRPO training and more stable gradient estimates—accelerating convergence without additional data or annotations.

## C Experiment Setting

Table 6 summarizes the key hyperparameters used in our experiments, including configurations for the GRPO optimization algorithm, the FLUX.1 text-to-image model, and the joint training process. For GRPO, we set the clipping range $\varepsilon$ to 0.2 and the KL penalty coefficient $\beta$ to 0.04 with a group size of 4. The FLUX.1 model operates at a resolution of $512 \times 512$, with 50 diffusion steps and a classifier-free guidance (CFG) scale of 3.5. In joint training, we balance ImageReward and VLM-Reward with equal weights (0.5 each), and constrain prompt lengths between 15 and 77 tokens. Training is conducted using 8 devices with a per-device batch size of 4, a learning rate of 2e-6, gradient accumulation steps of 2, and a total of 3 epochs.

## D Ablation Study

**Training Dynamics.** Figure 6 presents the reward curve during reinforcement learning of the Re-Prompt. We observe a stable and monotonically increasing trend in the reward, demonstrating that our reward model provides a reliable and effective supervision signal throughout training. The absence of sharp fluctuations or reward collapse suggests that our RL setup maintains stable policy updates. This aligns with the observed downstream improvements, confirming that reward-guided prompt refinement effectively enhances compositional alignment in generated images.

**Ablation Study on Visual Reasoning Rewards.** To investigate the effectiveness of different visual reasoning reward signals used in our method, we conduct a detailed ablation study on the GenEval benchmark. As shown in Table 7, our method consistently improves over the FLUX baseline and Qwen2.5 3B variant across all subcategories, demonstrating the efficacy of reward-driven learning in aligning generated images with prompt semantics. Both ImageReward and VLM-Reward show noticeable gains over the +Qwen2.5 3B baseline, indicating that each reward captures complementary aspects of visual faithfulness. However, their standalone performances are still limited in challenging tasks like Position. Notably, our ensemble reward formulation—which combines both ImageReward and VLM-based feedback—achieves the best overall performance, with consistent improvements compared to the +Qwen2.5 3B baseline. These are precisely the categories that require stronger compositional understanding and spatial reasoning, underscoring the strength of our composite reward design in driving semantically aligned image generation. The ablation confirms that each reward contributes uniquely, and their integration enables more holistic supervision, leading to substantial gains in compositional image-text alignment.

Table 6: Key Parameters for GRPO and T2I Model.

| Config | Symbol | Value |
|---|---|---|
| **GRPO Config** | | |
| Clipping range | $\varepsilon$ | 0.2 |
| KL penalty coefficient | $\beta$ | 0.04 |
| Group size | $G$ | 4 |
| **FLUX.1 Config** | | |
| Image resolution | $H \times W$ | $512 \times 512$ |
| Diffusion steps | $T$ | 50 |
| CFG scale | $\lambda_{\text{cfg}}$ | 3.5 |
| **Joint Training Parameters** | | |
| ImageReward weight | $\alpha$ | 0.5 |
| VLM-Reward weight | $\gamma$ | 0.5 |
| Min prompt's length | $L_{\min}$ | 15 |
| Max prompt's length | $L_{\max}$ | 77 |
| Device number | - | 8 |
| Per device batch size | $B$ | 4 |
| Learning rate | $lr$ | 2e-6 |
| Gradient Accumulation | - | 2 |
| Epoch | - | 3 |

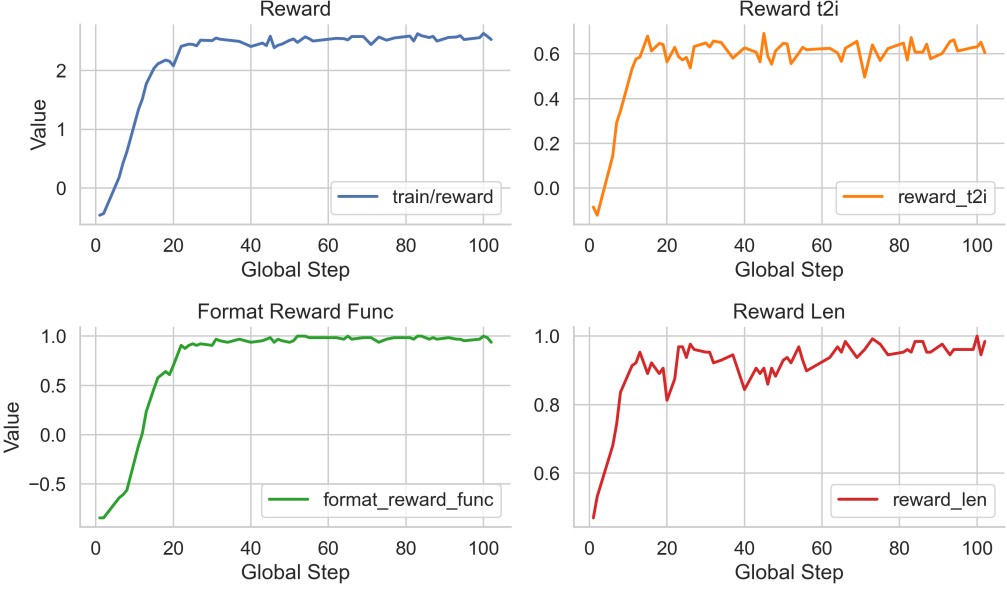

Figure 6: Training curve of our RePrompt during reinforcement learning. The curve shows steady reward improvement, indicating stable training dynamics and effective reward signal.

**Ablation Study on VLM-Reward Models.** To evaluate the generality and accessibility of our framework, we replaced GPT-4V with open-source vision-language models (Qwen-VL 72B and 32B) as the reward model. Results show that RePrompt still achieves competitive performance: 0.74 with Qwen-VL 72B and 0.73 with Qwen-VL 32B (vs. 0.76 with GPT-4V), confirming the method's robustness without reliance on proprietary tools. We will include full results and adoption with accessible reward models in the final version.

Table 7: **Ablation study of rewards on the GenEval benchmark.**

| Method | $\alpha$ | $\gamma$ | Single object | Two object | Counting | Colors | Position | Attribute binding | Overall ↑ |
|---|---|---|---|---|---|---|---|---|---|
| FLUX | - | - | 0.99 | 0.79 | 0.75 | 0.78 | 0.18 | 0.45 | 0.66 |
| +Qwen2.5 3B | - | - | 0.99 | 0.84 | 0.63 | 0.81 | 0.35 | 0.48 | 0.68 |
| **RePrompt** | 1.0 | 0.0 | 0.99 | 0.86 | 0.66 | 0.89 | 0.52 | 0.50 | 0.74 |
| | 0.7 | 0.3 | 0.98 | 0.86 | 0.70 | 0.88 | 0.56 | 0.52 | 0.75 |
| | 0.5 | 0.5 | 0.98 | 0.87 | 0.77 | 0.85 | 0.62 | 0.49 | 0.76 |
| | 0.3 | 0.7 | 0.99 | 0.85 | 0.74 | 0.84 | 0.58 | 0.50 | 0.75 |
| | 0.0 | 1.0 | 0.99 | 0.85 | 0.72 | 0.83 | 0.53 | 0.47 | 0.74 |

Table 8: Performance comparison of RePrompt with different VLM-Reward models

| Method | Single object | Two object | Counting | Colors | Position | Attribute binding | Overall ↑ |
|---|---|---|---|---|---|---|---|
| + GPT-4V | 0.98 | 0.87 | 0.77 | 0.85 | 0.62 | 0.49 | 0.76 |
| + Qwen2.5-VL 72B | 0.98 | 0.86 | 0.75 | 0.84 | 0.58 | 0.46 | 0.74 |
| + Qwen2.5-VL 32B | 0.98 | 0.85 | 0.74 | 0.83 | 0.55 | 0.45 | 0.73 |

**Training Dynamics with Different Visual Reasoning Rewards.** We analyze the impact of different visual reasoning rewards on training dynamics by plotting training curves under various reward configurations. As shown in Figure 7, training with a single reward model leads to unstable reward fluctuations and degraded sample quality. This instability arises from the limited supervision capacity of a single reward model, which may overfit to specific patterns or neglect important compositional aspects. In contrast, our proposed multi-reward formulation—balancing diverse reasoning signals, enables smoother optimization and more consistent improvements across iterations. These results emphasize the importance of combining complementary reward models to achieve both stability and generalization.

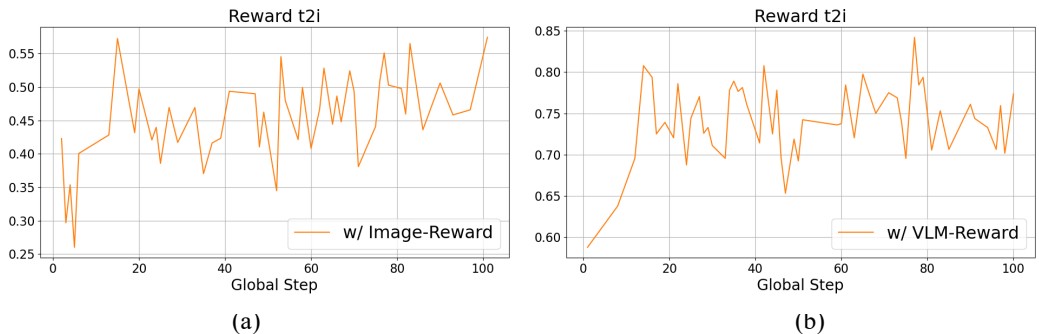

(a)  (b)

Figure 7: Training curve with different visual reasoning rewards. Using a single reward model leads to instability and suboptimal performance during training. In contrast, our balanced reward design—combining multiple specialized reward signals—yields more stable convergence and higher final reward values.

**Impact of Reasoning Length.** We investigated the impact of the reasoning length (maximum token limit for the CoT) on generation quality by comparing limits of 512, 1024, and 2048 tokens. As shown in Table 9, the 512-token setting yields the optimal performance with an Overall Score of 0.76. We observe a slight performance degradation as the length increases to 2048 (Overall: 0.73). Qualitative analysis suggests that excessively long reasoning chains may introduce hallucinations or dilute the core prompt instructions, making it harder for the T2I model to maintain focus. However, it is worth noting that even at 2048 tokens, our method still significantly outperforms the FLUX baseline (0.66), demonstrating the robustness of our approach across different context lengths.

Table 9: Ablation study on reasoning length (token limit). The 512-token setting achieves the best balance, while longer contexts (2048) show a slight decline but still outperform the baseline.

| Method | Single object | Two object | Counting | Colors | Position | Attribute binding | Overall ↑ |
|---|---|---|---|---|---|---|---|
| FLUX | **0.99** | 0.79 | 0.75 | 0.78 | 0.18 | 0.45 | 0.66 |
| Ours (512) | 0.98 | **0.87** | **0.77** | **0.85** | **0.62** | **0.49** | **0.76** |
| Ours (1024) | 0.98 | 0.85 | 0.76 | 0.83 | 0.56 | 0.48 | 0.74 |
| Ours (2048) | 0.97 | 0.83 | 0.75 | 0.81 | 0.52 | 0.47 | 0.73 |

Table 10: Human evaluation of reasoning traces (0-10 scale). High scores in Correctness and Consistency confirm that our method generates logically sound plans rather than just keywords. The slightly lower Conciseness reflects the necessary elaboration for T2I alignment.

| Method | Correctness | Consistency | Conciseness | Safety |
|---|---|---|---|---|
| GPT-5 | 9.9 | 9.7 | 7.9 | 9.9 |
| **Ours** | 9.7 | 9.5 | 8.3 | 9.8 |

**Evaluation of Reasoning Quality.** To address the concern regarding the quality and role of the reasoning traces, we conducted a human evaluation on the generated Chain-of-Thought (CoT) texts across four dimensions (0-10 scale). As shown in Table 10, our method achieves high scores in Correctness (9.7) and Consistency (9.5), comparable to the GPT-5. This demonstrates that the model generates logically sound and hallucination-free reasoning, confirming that the improvement is not merely due to keyword stuffing. While the RL process introduces model-preferred vocabulary—resulting in slightly lower Conciseness (8.3) due to necessary elaboration—this mechanism is essential for reducing generation stochasticity and enhancing controllability. Thus, the effectiveness stems from the synergy of structured reasoning ensuring logical coverage and RL-driven phrasing optimizing the final alignment.

## E    GENERALIZATION STUDY

**Generalization to Different Data Distributions.** We conducted a comprehensive retraining of our model using a completely new and diverse dataset, distinct from the *GenEval* benchmark distribution. Specifically, instead of relying on *GenEval*-like templates, we employed **Gemini 2.5 Pro** to simulate realistic user inputs derived from a large-scale collection of **100,000 diverse real-world images**. This procedure generated 100k supervised fine-tuning (SFT) samples and 10k reinforcement learning (RL) samples, reflecting a truly "in-the-wild" data distribution. We then evaluated the retrained model on the *GenEval* benchmark to measure its cross-distribution generalization. As shown in Table 11, our method yields significant performance gains compared to relevant baselines trained on the original datasets, demonstrating a clear improvement in all reasoning aspects. For example, on the *FLUX* benchmark, our model trained on the new distribution achieves an **Overall score of 0.75** versus 0.66 obtained by the FLUX baseline. This result strongly supports our claim that the model has learned generalized reasoning and instruction-following capabilities rather than overfitting to specific templates or benchmark patterns.

**Generalization across VLM Backbones.** To further validate the architectural generalization and scalability of our framework, we extended our training strategy to the **InternVL** family (specifically InternVL3.5-4B Wang et al. (2025b)). As presented in Table 12, our method proves to be **architecture-agnostic**, yielding consistent improvements across all T2I baselines regardless of the underlying VLM backbone. Notably, the InternVL-based model achieves superior performance compared to the Qwen2.5-3B version, boosting the Overall score on FLUX from 0.76 to **0.78**, with significant gains in challenging tasks like *Counting* (0.77 → 0.81) and *Position* (0.62 → 0.65). This demonstrates that RePrompt is not only robust to model selection but also effectively leverages stronger backbone capabilities to enhance instruction following.

Table 11: **Cross-Distribution Evaluation on GenEval.** The model was trained on a distinct "in-the-wild" dataset (generated by Gemini 2.5 Pro) to test generalization. Despite the distribution shift, RePrompt significantly improves performance across all backbones, particularly in spatial positioning and attribute binding.

| Method | Single object | Two object | Counting | Colors | Position | Attribute binding | Overall ↑ |
|---|---|---|---|---|---|---|---|
| FLUX (Base) | 0.99 | 0.79 | 0.75 | 0.78 | 0.18 | 0.45 | 0.66 |
| **+ Ours (Wild)** | 0.98 | **0.86** | 0.75 | **0.85** | **0.58** | **0.48** | **0.75** |
| SD3 (Base) | **1.00** | 0.85 | **0.62** | **0.88** | 0.22 | 0.58 | 0.69 |
| **+ Ours (Wild)** | 0.99 | 0.85 | 0.59 | 0.85 | **0.57** | **0.59** | **0.75** |
| PixArt-Σ (Base) | **0.99** | 0.60 | 0.47 | 0.81 | 0.10 | 0.26 | 0.54 |
| **+ Ours (Wild)** | 0.98 | **0.65** | **0.58** | **0.82** | **0.43** | **0.37** | **0.63** |

Table 12: **Generalization across VLM Backbones.** We compare our method trained with Qwen2.5-3B versus InternVL3.5-4B. The results show that our approach is architecture-agnostic and scales positively with stronger VLM backbones.

| Method | Single object | Two object | Counting | Colors | Position | Attribute binding | Overall ↑ |
|---|---|---|---|---|---|---|---|
| FLUX (Base) | 0.99 | 0.79 | 0.75 | 0.78 | 0.18 | 0.45 | 0.66 |
| + Ours (Qwen2.5-3B) | 0.98 | 0.87 | 0.77 | 0.85 | 0.62 | 0.49 | 0.76 |
| **+ Ours (InternVL3.5-4B)** | **0.99** | **0.89** | **0.81** | **0.86** | **0.65** | **0.52** | **0.78** |
| SD3 (Base) | **1.00** | 0.85 | 0.62 | **0.88** | 0.22 | 0.58 | 0.69 |
| + Ours (Qwen2.5-3B) | 0.99 | 0.86 | 0.60 | 0.86 | 0.59 | 0.60 | 0.75 |
| **+ Ours (InternVL3.5-4B)** | **1.00** | **0.87** | **0.62** | 0.87 | **0.61** | **0.62** | **0.76** |
| PixArt-Σ (Base) | **0.99** | 0.60 | 0.47 | 0.81 | 0.10 | 0.26 | 0.54 |
| + Ours (Qwen2.5-3B) | 0.98 | 0.64 | 0.56 | 0.81 | 0.40 | 0.35 | 0.62 |
| **+ Ours (InternVL3.5-4B)** | **0.99** | **0.66** | **0.58** | **0.82** | **0.42** | **0.37** | **0.64** |

**Generalization on Diverse and Long-form Narratives.** To demonstrate RePrompt's capability on diverse and long-form narrative prompts, we evaluated its generalizability on the TIIF-bench Wei et al. (2025), a comprehensive benchmark that categorizes prompts by length (Short vs. Long) and complexity (Basic vs. Advanced). As shown in Table 13, our method demonstrates robust generalizability across multiple backbones. Crucially, addressing the concern about long-form narrative and complex scenarios, our method achieves significant improvements in the Long and Advanced categories. For instance, with FLUX, we observe a substantial gain of nearly 8 points on Advanced-Long prompts. This result suggests that our reasoning-augmented framework effectively manages the higher cognitive load required for long-context descriptions, ensuring that complex narrative elements are structurally planned and preserved rather than being lost during generation.

**Generalization across Architectures.** To assess the versatility of our method beyond diffusion-based frameworks, we conducted experiments on BAGEL Deng et al. (2025) (a unified multimodal model) and Infinity Han et al. (2024) (an autoregressive model). As detailed in Table 14, our method yields substantial improvements across both architectures. Specifically, for BAGEL, the Overall score increases from 0.79 to 0.86, with the most notable gain in Position (+0.15), indicating significantly better spatial control. For Infinity, we observe a marked boost in Counting (+0.13) and Position (+0.16), driving the Overall score from 0.69 to 0.77. These results confirm that our RL-driven prompt optimization effectively adapts to the distinct decoding mechanisms of autoregressive and unified models. By tailoring the reasoning strategy to the specific latent characteristics of each backbone, our method successfully mitigates their inherent weaknesses in spatial and numerical reasoning, justifying the use of model-specific policy alignment.

**Human Evaluation.** To validate the practical utility and interpretability of our method beyond automatic metrics, we conducted a human user study with 15 evaluators on 30 randomly selected

Table 13: Performance comparison on TIIF-bench. Our method consistently enhances generation quality, particularly for long and advanced prompts.

| Model | Overall | | Basic | | Advanced | |
|---|---|---|---|---|---|---|
| | Short | Long | Short | Long | Short | Long |
| FLUX | 71.09 | 71.78 | 83.12 | 78.65 | 65.79 | 68.54 |
| + Ours | **76.50** | **77.80** | **88.50** | **84.20** | **72.80** | **76.50** |
| SD 3 | 67.46 | 66.09 | 78.32 | 77.75 | 61.46 | 59.56 |
| + Ours | **72.10** | **71.50** | **83.50** | **82.00** | **67.50** | **66.80** |
| PixArt-$\Sigma$ | 62.00 | 58.12 | 70.66 | 75.25 | 57.65 | 49.50 |
| + Ours | **67.80** | **65.50** | **76.50** | **80.10** | **64.20** | **58.00** |

Table 14: Generalization on architecturally diverse models (Unified Multimodal and Autoregressive). Our method significantly improves performance on both BAGEL and Infinity, demonstrating effectiveness beyond diffusion-based architectures.

| Method | Single object | Two object | Counting | Colors | Position | Attribute binding | Overall ↑ |
|---|---|---|---|---|---|---|---|
| BAGEL | 0.98 | 0.90 | 0.79 | 0.85 | 0.62 | 0.60 | 0.79 |
| + Ours | **0.98** | **0.94** | **0.82** | **0.92** | **0.75** | **0.75** | **0.86** |
| Infinity | 0.97 | 0.85 | 0.45 | 0.85 | 0.54 | 0.48 | 0.69 |
| + Ours | **0.99** | **0.91** | **0.58** | **0.90** | **0.70** | **0.54** | **0.77** |

examples. Evaluators rated samples on a scale of 1-5 across three dimensions: Text-Image Alignment, Image Quality, and Prompt Readability. As presented in Table 15, our method consistently outperforms both the original baselines and the Qwen2.5-enhanced versions. Notably, for FLUX, we achieve a Text-Image Alignment score of 4.48 (vs. 3.72 for the baseline). Crucially, addressing the concern regarding the nature of the generated text, our method maintains high Prompt Readability scores. This confirms that the RL optimization guides the model to produce coherent, human-understandable reasoning and prompts, rather than degrading into unreadable adversarial tokens.

# F   MORE CASES

**More Qualitative Comparison.**   Figure 10, 11, 12 and 13 present qualitative comparisons on different types of compositional prompts, including Position, Two-object, Color, and Attribute Binding. Compared to baseline methods such as DELL-E3, our approach produces images that more faithfully adhere to the spatial, numerical, and attribute-based constraints specified in the prompts. For instance, in Figure 7, our method accurately grounds relative positions (e.g., "above", "left of") by leveraging explicit reasoning, while DELL-E3 often fails to reflect such relations or misplaces objects entirely. Figure 8 highlights our advantage in handling prompts involving multiple objects, where baseline models tend to merge objects or hallucinate irrelevant content. Similarly, in Figure 9, our method is better at preserving specified colors for each object, whereas the baseline sometimes misbinds colors or applies them inconsistently. Finally, as shown in Figure 10, our approach improves attribute binding, ensuring that each attribute is applied to the correct object without confusion. These results demonstrate the effectiveness of integrating reasoning to improve alignment between visual outputs and complex prompt semantics.

**Qualitative Results on Open-World Prompts.**   To verify the generalizability of RePrompt beyond standard object-centric benchmarks, we present qualitative results on open-world prompts in Figure 8. These prompts involve abstract concepts, specific artistic styles, and imaginative narratives, which pose significant challenges for base models due to their lack of concrete visual mappings. As observed, RePrompt effectively bridges the gap between abstract user intent and concrete visual attributes. By leveraging Chain-of-Thought reasoning, our method decomposes high-level

Table 15: Human evaluation results (1-5 scale). Our method significantly improves alignment and quality while maintaining high prompt readability, confirming that the generated reasoning remains coherent and human-understandable.

| Method | Text-Image Alignment | Image Quality | Prompt Readability |
|---|---|---|---|
| FLUX | 3.72 | 4.45 | 3.50 |
| FLUX + Qwen2.5 3B | 3.95 | 4.48 | 4.25 |
| FLUX + **Ours** | **4.48** | **4.55** | **4.38** |
| SD 3 | 3.85 | 4.38 | 3.60 |
| SD 3 + Qwen2.5 3B | 3.88 | 4.40 | 4.22 |
| SD 3 + **Ours** | **4.32** | **4.46** | **4.35** |
| PixArt-$\Sigma$ | 3.15 | 3.65 | 3.45 |
| PixArt-$\Sigma$ + Qwen2.5 3B | 3.48 | 3.78 | 4.15 |
| PixArt-$\Sigma$ + **Ours** | **3.85** | **3.92** | **4.28** |

stylistic and narrative descriptions into precise visual cues. Consequently, the generated images not only adhere to the complex constraints but also maintain high aesthetic quality, demonstrating that RePrompt is robust for diverse, real-world creative applications.

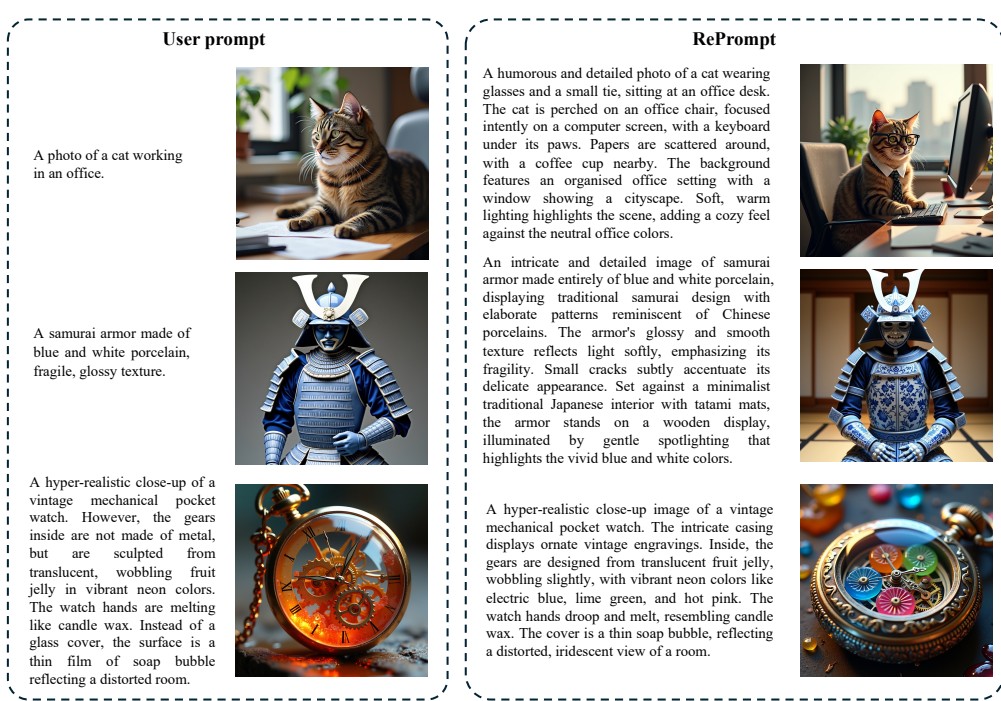

Figure 8: Qualitative Results on Open-World Prompts. We showcase RePrompt's performance on diverse, imaginative, and stylized prompts. The results demonstrate that our method effectively handles complex narratives and abstract concepts beyond simple object composition.

**Failure Case Analysis.** While RePrompt significantly enhances prompt adherence, we observe two primary failure modes through qualitative analysis as shown in Figure 9. The first is hallucination via over-reasoning, where the model generates an excessively verbose reasoning trace. This can lead to over-specification, introducing specific constraints not present in the original user input. The second stems from intrinsic base model limitations. In scenarios involving extreme spatial overlaps, RePrompt may generate a precise and correct prompt, yet the underlying T2I model fails to render the visual attributes correctly due to its own capacity constraints.

| User prompt | RePrompt |
|---|---|
| A photo of a bus above a boat. 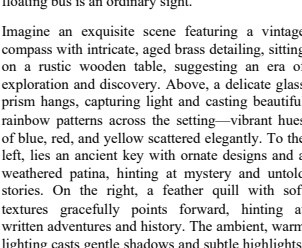 | In a serene, shimmering bay, a modern bus enigmatically floats above a luxurious yacht. At first glance, the bus seems to be supported by invisible structures, but as you look closer, those supports dissolve into the air, leaving the bus magically suspended. The water beneath the yacht is mirror-like, perfectly reflecting this bizarre phenomenon and amplifying its surreal effect. The bus shimmers subtly in the sunlight, its edges occasionally blurring and sending ripples through the air like a mirage. Passengers on the yacht, unfazed, continue their leisurely activities, as if the floating bus is an ordinary sight. 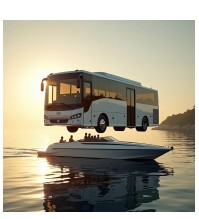 |
| A vintage compass rests on the table. Above it, a delicate glass prism hangs, casting rainbows. To the left lies an ancient key, while a feather quill points to the right. | Imagine an exquisite scene featuring a vintage compass with intricate, aged brass detailing, sitting on a rustic wooden table, suggesting an era of exploration and discovery. Above, a delicate glass prism hangs, capturing light and casting beautiful rainbow patterns across the setting—vibrant hues of blue, red, and yellow scattered elegantly. To the left, lies an ancient key with ornate designs and a weathered patina, hinting at mystery and untold stories. On the right, a feather quill with soft textures gracefully points forward, hinting at written adventures and history. The ambient, warm lighting casts gentle shadows and subtle highlights, enriching the historical and adventurous feel of the tableau 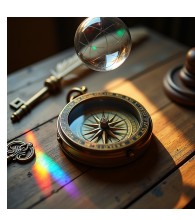 |

Figure 9: Visualization of Failure Cases. We observe two main failure patterns: (a) Over-Reasoning, where the model hallucinates unrequested details due to excessive interpretation; and (b) Base Model Limitation, where the T2I model fails to render complex attributes despite receiving a correct prompt.

## G    LIMITATIONS

While our method consistently improves compositional quality across standard T2I benchmarks, several limitations warrant future exploration. First, the performance gains on certain fine-grained tasks—such as numeracy and object counting—remain modest, suggesting potential for further enhancement in precise quantitative reasoning. Second, our approach requires fine-tuning with compositional supervision, which may introduce additional computational cost and reliance on structured training signals. However, this design is consistent with common RLHF setups and does not limit practical deployment. Third, the effectiveness of RePrompt depends on the quality of the reward model; while we demonstrate robustness across standard evaluators, improvements in reward fidelity could further amplify performance.

## H    LLM USAGE

In this work, large language models (LLMs) were utilized primarily as a general-purpose assistive tool to help with text polishing and improving writing clarity. The core research ideation, experimental design, analysis, and conclusions were developed solely by the authors. The LLM did not contribute to generating any original scientific content or ideas. The authors take full responsibility for the contents of the paper, including all text generated or revised with the assistance of LLMs.

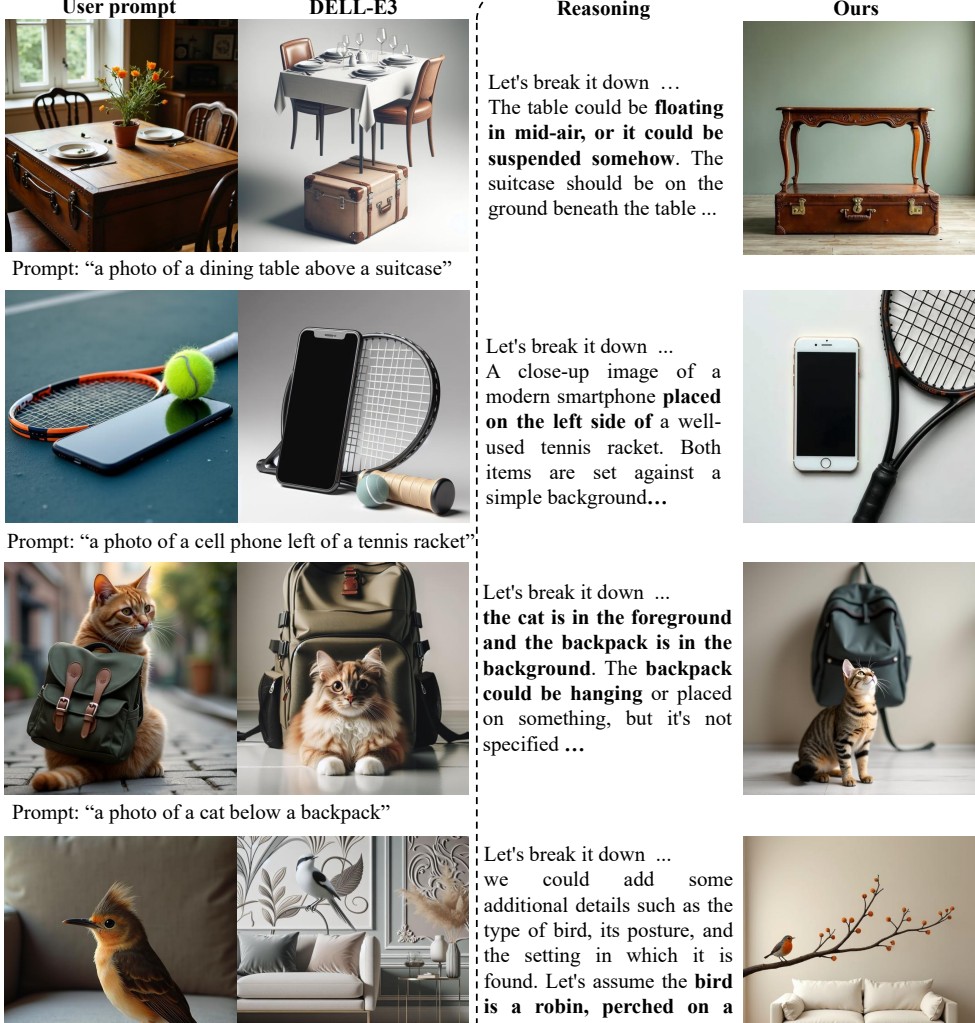

Figure 10: Qualitative results on compositional prompts (Position). We compare DELL-E3, the intermediate reasoning process, and our final RePrompt outputs. While DELL-E3 often struggles with spatial relations (e.g., object positions), our reasoning-guided approach enables more accurate and faithful generations that better match the prompt's compositional constraints.

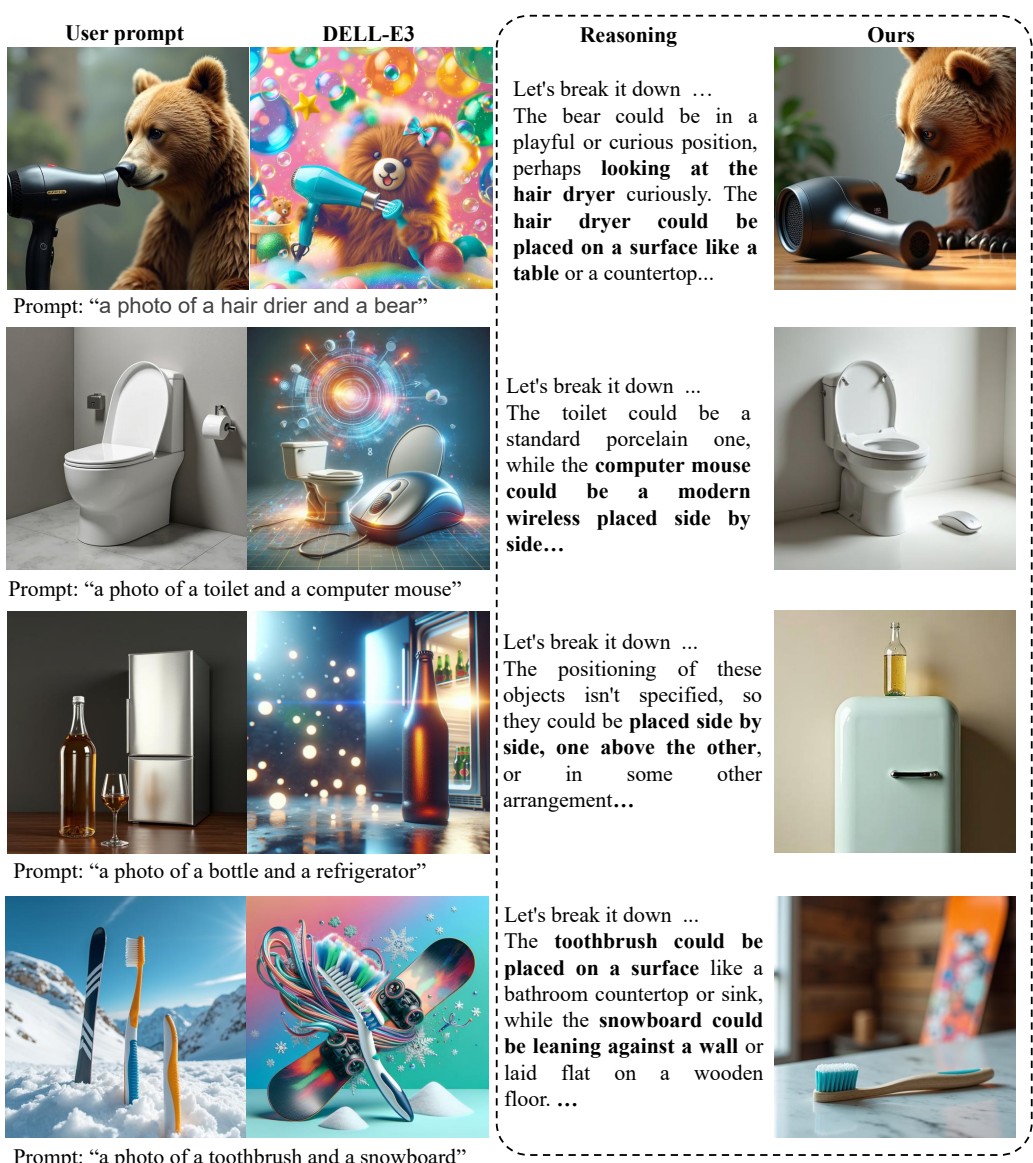

Figure 11: Qualitative results on compositional prompts (Two-object). We show comparisons among DELL-E3, our reasoning process, and RePrompt outputs on prompts involving two objects. DELL-E3 often fails to generate both entities accurately, whereas our method uses explicit reasoning to guide the model in generating semantically correct and compositionally faithful images.

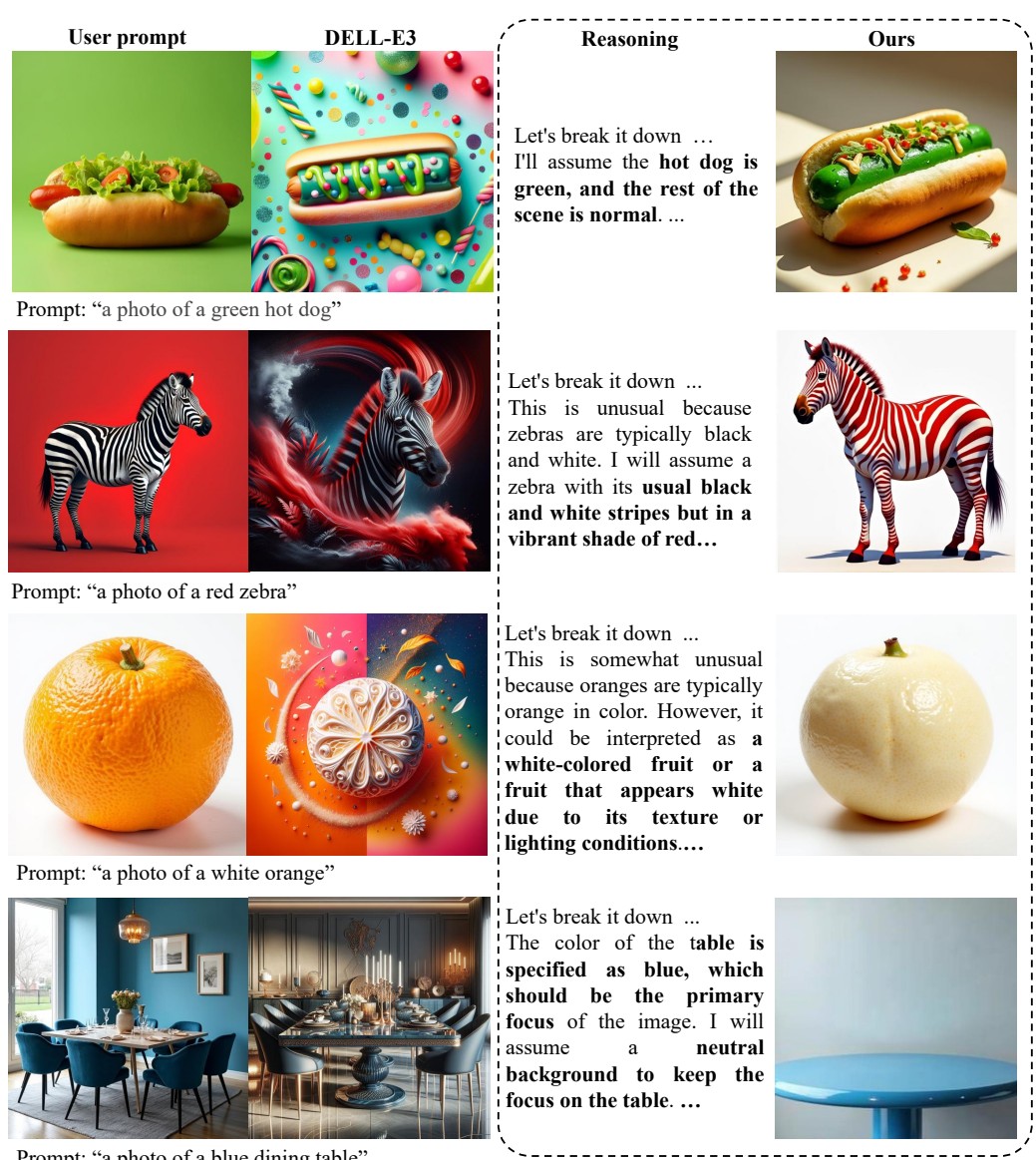

Figure 12: Qualitative results on compositional prompts (Color). We present qualitative comparisons on prompts involving specific color attributes. While DELL-E3 tends to ignore or misinterpret color constraints, our approach leverages explicit reasoning to ensure accurate color grounding for each object, resulting in more faithful visual compositions.

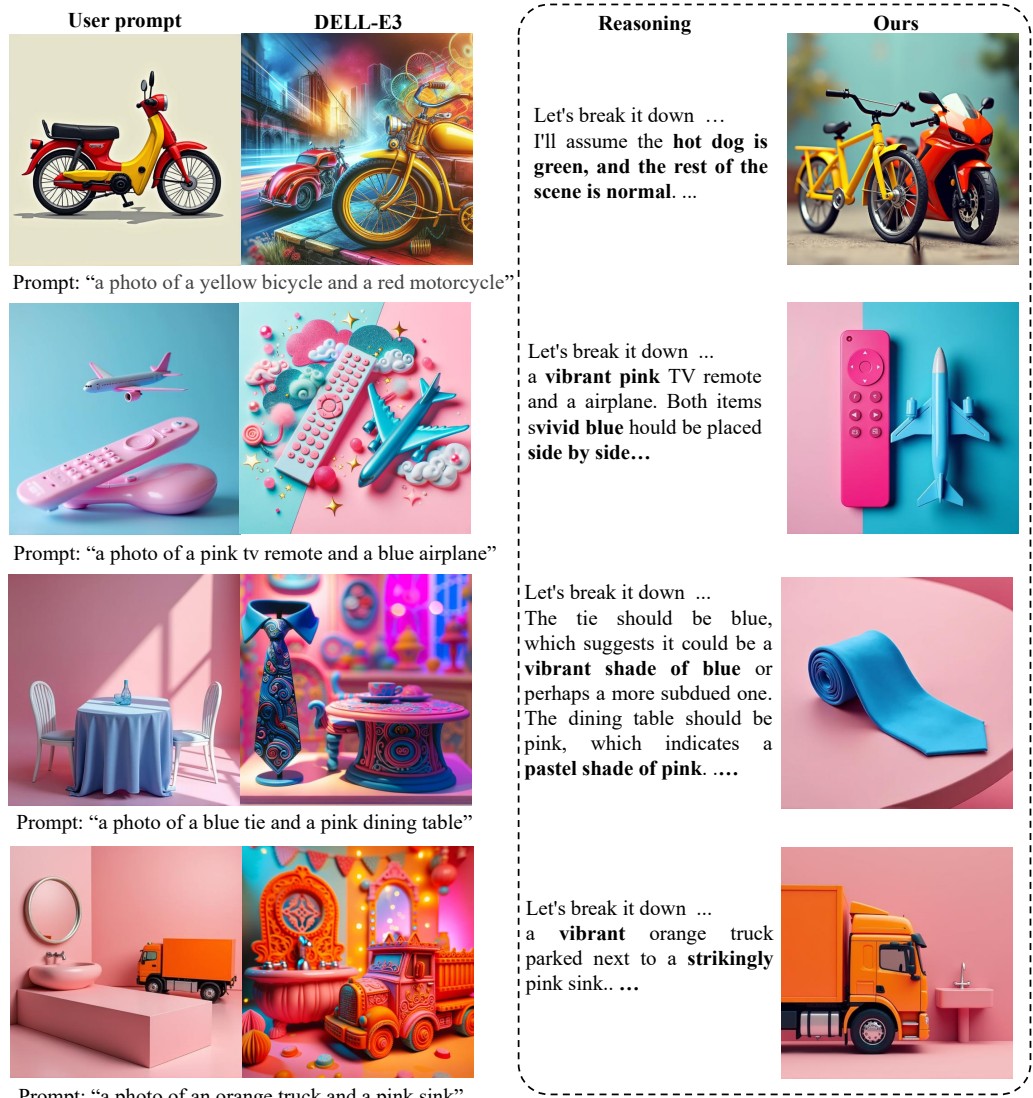

Figure 13: Qualitative results on compositional prompts (Attribute binding). We show examples where prompts specify multiple attributes that must be correctly associated with the corresponding objects. Our method utilizes step-by-step reasoning to disambiguate attribute-object bindings, avoiding attribute swaps or omissions that are common in baseline outputs.

