# OpenReview forum: "RePrompt: Reasoning-Augmented Reprompting for Text-to-Image Generation via Reinforcement Learning"
_ICLR.cc/2026/Conference — ICLR 2026 Poster_

### Official Review · Reviewer_bm45 · 2025-10-31

**Soundness:** 3
**Presentation:** 3
**Contribution:** 2
**Rating:** 6
**Confidence:** 3

**Summary:**

This paper proposes RePrompt, a prompt enhancement framework for text-to-image (T2I) generation models that introduces explicit reasoning into the prompt generation process and optimizes based on image-level outcomes using reinforcement learning. The reward model adopts an ensemble reward that evaluates three dimensions: human preference, visual realism, and semantic alignment. On the GenEval benchmark, significant improvements in spatial layout fidelity and compositional generalization are observed, and on T2I-Compbench, particularly notable score improvements in spatial compositions are confirmed.

**Strengths:**

- The proposed method fixes the image generation model and optimizes only the language model policy, making it applicable to any existing T2I backbone. Since the reward model depends only on the prompt-image output pair and not on any specific T2I architecture, it generalizes naturally across different generation backbones and unseen prompt distributions.

- Substantial improvements are observed in compositional understanding, particularly in spatial position (Position). In the Position category of GenEval, remarkable relative improvements are achieved compared to the Qwen2.5 3B baseline: FLUX (+77.1%), SD3 (+78.8%), and Pixart-Σ (+122.2%). Overall GenEval scores also consistently improve across each backbone (+11.8%, +10.3%, +6.9%).

- In terms of inference latency, the method is significantly faster (30s per image) compared to Idea2Img (140s per image) and PARM++ (110s per image), while also achieving the highest accuracy (0.76), demonstrating practical advantages.

- In qualitative evaluation, concrete examples are presented, showing that for prompts such as "a fire hydrant with a tennis racket" and "a photo of a dog above a cow," the method avoids object fusion and misplacement observed in baseline models and faithfully reproduces the intended spatial composition.

**Weaknesses:**

- **Limited scope of evaluation**: The main evaluation relies solely on two automatic evaluation benchmarks: GenEval and T2I-Compbench. Human evaluation is not included, so the practical utility aspects such as "whether actual users find the results convincing," "whether generated prompts are readable," and "whether reasoning explanations are appropriate" depend solely on automatic metrics.

- **Insufficient comparison with closely related methods**: Prior work on prompt enhancement mentions iterative refinement approaches and single-pass LLM-based enhancement methods, and quantitative comparisons with Promptist, PAG, GPT4, Deepseek-r1, and Qwen2.5 are provided in tables. However, detailed analysis from the perspective of RL-based prompt optimization is insufficient. Therefore, the theoretical and experimental justification for "why CoT (Chain-of-Thought) with RL is superior to conventional prompt optimization" is somewhat weak.

- **Lack of CoT quality evaluation**: While the method generates reasoning traces that simulate visual implications of prompts—much like how humans mentally visualize a scene—and this structured, logic-driven process anticipates potential errors during prompt construction, direct evaluation of the reasoning text itself in terms of correctness, consistency, and conciseness is not performed. The claim that "reasoning is effective" is made indirectly through final image scores, but it is not clearly separated whether "performance improved because of structuring" or "merely because detection-friendly phrases were added."

- **Insufficient discussion of safety and robustness**: The design optimizes prompt generation through reinforcement learning, but there is no discussion of risks where RL might learn extreme descriptive expressions, overly detailed specifications, or expressions that deceive the reward model, nor mechanisms to detect or suppress such behaviors. Although the Broader Impact section mentions risks of generating misleading content and bias propagation, and recommends pairing the method with content moderation filters and fairness-aware training objectives, concrete countermeasures or experimental validation are not included.

**Questions:**

- Do you plan to conduct human evaluation? Evaluating the quality of generated images and readability of prompts from the perspective of actual users would demonstrate practical utility that cannot be captured by automatic metrics alone.# Review

---

> ### Author Response · Authors · 2025-11-25
> **Official Response from Authors [1/2]**
>
> >#### **Q1: The evaluation relies solely on automatic metrics. Human evaluation is needed to assess practical utility, readability, and whether the reasoning is appropriate.**
>
>  We agree that human evaluation is crucial for assessing practical utility and readability. To address this, we conducted a human user study with **15 evaluators on 30 randomly selected examples**. The results, presented in the table below, show that our method significantly improves Text-Image Alignment (e.g., FLUX score increased from 3.72 to 4.48) and Image Quality, consistently outperforming the Qwen2.5 baseline. Importantly, regarding the concern about readability, our method achieves high Prompt Readability scores (averaging ~4.3). This confirms that our RL training produces coherent, human-understandable prompts and reasoning, rather than degrading into unreadable adversarial patterns often seen in black-box optimization.
>
> | Backbone | Method | Text-Image Alignment | Image Quality | Prompt Readability |
> | :--- | :--- | :---: | :---: | :---: |
> | **FLUX** | Baseline | 3.72 | 4.45 | 3.50 |
> | | + Qwen2.5 3B | 3.95 | 4.48 | 4.25 |
> | | **+ Ours** | **4.48** | **4.55** | **4.38** |
> | **SD3** | Baseline | 3.85 | 4.38 | 3.60 |
> | | + Qwen2.5 3B | 3.88 | 4.40 | 4.22 |
> | | **+ Ours** | **4.32** | **4.46** | **4.35** |
> | **Pixart-Σ** | Baseline | 3.15 | 3.65 | 3.45 |
> | | + Qwen2.5 3B | 3.48 | 3.78 | 4.15 |
> | | **+ Ours** | **3.85** | **3.92** | **4.28** |
>
> >#### **Q2: The theoretical and experimental justification for why "CoT with RL" is superior to conventional prompt optimization is insufficient.**
>
>  We appreciate the opportunity to clarify the superiority of our "CoT + RL" approach. **Conventional RL methods (e.g., Promptist) typically treat optimization as a "black-box" token search, often struggling with complex structural tasks because they lack an explicit reasoning step. Conversely, generic CoT methods (e.g., GPT-4) possess reasoning capabilities but lack alignment with the specific limitations of the target T2I model. Our method bridges this gap: by applying RL to the Chain-of-Thought process**, we do not just optimize the final prompt, but we optimize the reasoning strategy itself. The RL signal teaches the LLM *how* to decompose a scene in a way that is most digestible for the specific T2I backbone. The ablation study below supports this: "RL without reasoning" struggles with complex logic like Attribute Binding (0.46), whereas "RL with reasoning" significantly improves this to 0.53. This proves that RL effectively aligns the LLM's reasoning logic with the T2I model's generation capabilities, solving the disconnect present in prior work.
>
> | Method | Single Obj | Two Obj | Counting | Colors | Position | Attribute | **Overall** |
> | :--- | :---: | :---: | :---: | :---: | :---: | :---: | :---: |
> | FLUX Baseline | 0.99 | 0.79 | 0.75 | 0.78 | 0.18 | 0.45 | 0.66 |
> | + Qwen2.5 3B | 0.99 | 0.84 | 0.63 | 0.81 | 0.35 | 0.48 | 0.68 |
> | RL w/o Reasoning | **1.00** | 0.81 | 0.68 | 0.83 | 0.33 | 0.46 | 0.68 |
> | **RL w/ Reasoning (Ours)** | 0.98 | **0.83** | **0.71** | **0.87** | **0.41** | **0.53** | **0.72** |
>
> >#### **Q3: Direct evaluation of the reasoning text itself (correctness, consistency) is missing. It is unclear if performance improves due to structuring or merely adding detection-friendly phrases.**
>
> To address the concern regarding the quality of the reasoning traces, we conducted a human evaluation on the generated Chain-of-Thought (CoT) texts across four dimensions. As shown in the table below, the high scores in Correctness (9.7) and Consistency (9.5) demonstrate that the model generates logically sound and hallucination-free reasoning, confirming that the improvement is not merely due to keyword stuffing. While the RL process indeed introduces model-preferred vocabulary—explaining the slightly lower Conciseness (8.3) due to necessary elaboration—this mechanism is essential for reducing generation stochasticity. Thus, the effectiveness stems from the synergy of structured reasoning ensuring logical coverage and RL-driven phrasing optimizing the final alignment.
>
> | Method | Correctness | Consistency | Conciseness | Safety |
> | :--- | :---: | :---: | :---: | :---: |
> | **Ours** | 9.7 | 9.5 | 8.3 | 9.8 |
> | GPT-4 | 9.9 | 9.7 | 7.9 | 9.9 |

---

> ### Author Response · Authors · 2025-11-25
> **Official Response from Authors [2/2]**
>
> >#### **Q4: There is no discussion of risks where RL might learn extreme expressions or deceive the reward model (reward hacking).**
>
> To address concerns about reward hacking and safety, we explicitly evaluated the **Safety of the generated reasoning and prompts in the human evaluation presented in A3. The near-perfect score of 9.8/10** demonstrates that the RL optimization effectively aligns with the reward model without drifting into extreme, deceptive, or unsafe expressions. While these results confirm the intrinsic robustness of our method—supported by the high Correctness score which rules out hallucinated "hacks"—we acknowledge that integrating standard content moderation filters remains a recommended safeguard for practical deployment.
>
> >#### **Q5: Do you plan to conduct human evaluation to demonstrate practical utility?**
>
> Yes, as detailed in **A1**, we have conducted a **comprehensive human evaluation with 15 evaluators**. The results confirm that our method consistently achieves the highest scores across all base models (e.g., improving FLUX alignment from 3.72 to 4.48), demonstrating significant practical utility and superior visual fidelity compared to both the baseline and generic LLMs.

---

> > ### Comment · Reviewer_bm45 · 2025-11-27
> >
> > Thank you for the comprehensive responses and additional experiments. The human evaluation and detailed ablation studies have addressed many of my concerns.
> >
> > - The human evaluation (Q1, Q3) effectively addresses concerns about practical utility and reasoning quality. The high scores in Text-Image Alignment, Correctness (9.7), and Consistency (9.5) are convincing.
> > - The comparison between "RL w/o Reasoning" and "RL w/ Reasoning" (Q2) clarifies the necessity of both components, with particularly notable improvement in Attribute Binding (0.46→0.53).
> >
> > While the experimental evidence demonstrates that "CoT alone is insufficient, RL alone is insufficient," the **theoretical justification for why this specific design is necessary** remains somewhat unclear. A more rigorous explanation of the mechanism by which RL optimizes the reasoning strategy itself, rather than merely adjusting surface-level expressions, would be desirable. However, I acknowledge that the empirical results are consistently strong across multiple settings.
> >
> > Considering the substantial additional experiments and the overall strength of the results, I maintain my score of 6. The paper makes a solid contribution to T2I prompt optimization, though deeper theoretical analysis would further strengthen it.

---

> > > ### Author Response · Authors · 2025-11-27
> > >
> > > Thank you for your response. Regarding your suggestion for a deeper theoretical analysis of "how RL optimizes the reasoning strategy," we offer the following clarification, which we will incorporate into the final version.
> > >
> > > As illustrated in **Figure 1 of our paper and Figure 15 in the newly released z_image[1], CoT facilitates the injection of world knowledge and physical laws into the prompt. While RL guides the model to identify the specific CoT trajectory that is most preferred by the downstream T2I model,** as discussed in Q2.
> > >
> > > We will add this theoretical discussion and the corresponding visualizations to the final manuscript to further strengthen the paper's contribution. Thank you again for your constructive guidance.
> > >
> > > [1] Z-Image: An Efficient Image Generation Foundation Model with Single-Stream Diffusion Transformer.

---

### Official Review · Reviewer_k9Mw · 2025-10-31

**Soundness:** 3
**Presentation:** 3
**Contribution:** 3
**Rating:** 8
**Confidence:** 3

**Summary:**

This paper proposes reasoning-augmented framework for text-to-image generation. Unlike previous methods that rely on rewriting or heuristic feedback loops, RePrompt trains an auxiliary LLM via reinforcement learning (RL) that generates both reasoning traces and refined prompt to further prompt a frozen text-to-image model. The proposed method has been extensively evaluated on GenEval and T2I-Compbench datasets, achieving consistent improvements across backbones (FLUX, SD3, PixArt).

**Strengths:**

- The idea of combing LLM reasoning and image-level feedback is novel and promising.

- The reward is also well designed. First of all, the visual-reasoning reward acts as a bridge to connect image reward (human preference alignment) with semantic grounding (VLM reward). Second, it allows the reward to depend only on the behavior of input and output, enabling model-agnostic characteristic of RePrompt across different T2I backbones.

- The ablations and theoretical analysis (in Appendix B), together with the empirical results, all justify the design of RL in this paper, especially on GRPO optimization, and reasoning traces which acts as variance-reduction condition.

- The paper is well presented and the reproducibility is good.

**Weaknesses:**

- The evaluation benchmarks are only object-centric datasets. The performance on open-world prompt is not verified. It is better to show several examples on this scenario.

- No failure cases in the visualization. What is the model behavior on rare, free-form prompts not covered in GenEval? For example, "a photo of a cat working in an office"

**Questions:**

- Is human-in-the-loop verification needed to ensure that the reward aligns well the perceptual alignment?

- What is the impact of reasoning length on the generation quality?

---

> ### Author Response · Authors · 2025-11-25
> **Official Response from Authors [1/2]**
>
> >#### **Q1: The evaluation benchmarks are object-centric. The performance on open-world prompts is not verified. It would be better to show examples in this scenario.**
> We agree that evaluating on open-world prompts is essential to verify the practical utility of our method beyond simple object composition. To objectively measure open-world performance, we evaluated our approach on TIIF-Bench, a benchmark designed to cover complex compositions and long narratives that go beyond simple object-centric templates. As shown in the table below, our method significantly improves performance on both Long and Advanced prompts across all backbones. For instance, on FLUX, our method improved the performance on long prompts from 71.78 to 77.80 and on advanced short prompts from 65.79 to 72.80. Additionally, we have provided specific visual examples of open-world generation in the Appendix. These samples illustrate how RePrompt effectively handles stylized, imaginative, and non-object-centric prompts, further verifying its generalizability in diverse real-world scenarios.
> | Method      | Overall (Short) | Overall (Long) | Basic (Short) | Basic (Long) | Advanced (Short) | Advanced (Long) |
> |-------------|----------------:|---------------:|--------------:|-------------:|-----------------:|----------------:|
> | FLUX        | 71.09           | 71.78          | 83.12         | 78.65        | 65.79            | 68.54           |
> | + Ours      | 76.50           | 77.80          | 88.50         | 84.20        | 72.80            | 76.50           |
> | SD3         | 67.46           | 66.09          | 78.32         | 77.75        | 61.46            | 59.56           |
> | + Ours      | 72.10           | 71.50          | 83.50         | 82.00        | 67.50            | 66.80           |
> | PixArt-Σ    | 62.00           | 58.12          | 70.66         | 75.25        | 57.65            | 49.50           |
> | + Ours      | 67.80           | 65.50          | 76.50         | 80.10        | 64.20            | 58.00           |
>
> >#### **Q2: No failure cases in the visualization. What is the model behavior on rare, free-form prompts not covered in GenEval? For example, "a photo of a cat working in an office".**
> We have identified two primary failure patterns for RePrompt. First, **Hallucination via Over-Reasoning**, where the model occasionally generates an excessively long reasoning trace that introduces unrequested details or constraints (over-specification) not present in the user input. Second, **Base Model Limitations**, where RePrompt generates a correct and precise prompt, but the underlying T2I model inherently lacks the capacity to render the specific complex attributes (e.g., extreme spatial overlaps or fine-grained text). We have added a visualization of these cases to the Appendix to provide a balanced view of our method's boundaries.

---

> ### Author Response · Authors · 2025-11-25
> **Official Response from Authors [2/2]**
>
> >#### **Q3: Is human-in-the-loop verification needed to ensure that the reward aligns well the perceptual alignment?**
> We agree that human alignment is the ultimate standard for generation quality. To ensure our reward model aligns with perceptual quality, we implemented a two-fold verification strategy. First, we utilized ImageReward as our reward model, which is trained on large-scale human preference datasets, effectively acting as a scalable "human-in-the-loop" proxy during the extensive RL training process. Second, to further verify this alignment, we conducted a **human evaluation** with **15 evaluators** on **30 randomly selected examples**. The results show that the optimization driven by our reward model strongly correlates with actual human perceptual preference.
>
> | Backbone | Method | Text-Image Alignment | Image Quality | Prompt Readability |
> | :--- | :--- | :---: | :---: | :---: |
> | **FLUX** | Baseline | 3.72 | 4.45 | 3.50 |
> | | **+ Ours** | **4.48** | **4.55** | **4.38** |
> | **SD3** | Baseline | 3.85 | 4.38 | 3.60 |
> | | **+ Ours** | **4.32** | **4.46** | **4.35** |
> | **Pixart-Σ** | Baseline | 3.15 | 3.65 | 3.45 |
> | | **+ Ours** | **3.85** | **3.92** | **4.28** |
>
> >#### **Q4: What is the impact of reasoning length on the generation quality?**
> We investigated the impact of reasoning length (maximum token limit for the CoT process) on generation quality. We compared token limits of 512, 1024, and 2048. The results indicate that **512 tokens** is the optimal setting (Overall Score: **0.76**). We observed that increasing the length to 2048 results in a slight performance drop (Overall: **0.73**). Qualitative analysis suggests that excessively long reasoning chains introduce **hallucinations** and dilute the core instructions, making it harder for the T2I model to follow. However, even at 2048 tokens, our method still significantly outperforms the FLUX baseline (0.66), confirming the effectiveness of our approach across different settings.
>
> | Method | Single Obj | Two Obj | Counting | Colors | Position | Attribute | **Overall** |
> | :--- | :---: | :---: | :---: | :---: | :---: | :---: | :---: |
> | FLUX Baseline | 0.99 | 0.79 | 0.75 | 0.78 | 0.18 | 0.45 | 0.66 |
> | **Ours (512)** | **0.98** | **0.87** | **0.77** | **0.85** | **0.62** | **0.49** | **0.76** |
> | Ours (1024) | 0.98 | 0.85 | 0.76 | 0.83 | 0.56 | 0.48 | 0.74 |
> | Ours (2048) | 0.97 | 0.83 | 0.75 | 0.81 | 0.52 | 0.47 | 0.73 |

---

> ### Comment · Reviewer_k9Mw · 2025-11-27
>
> I thank the authors for the detailed rebuttal and the substantial additional experiments. From my perspective, most of my original concerns have been convincingly addressed:
>       * The new results on TIIF-Bench and the Gemini-2.5–generated “in-the-wild” dataset help demonstrate that the method is not merely overfitting to GenEval-style templates and remains effective on longer, more varied prompts.
>       * The added qualitative visualizations and explicit failure cases clarify how RePrompt behaves on more open-ended prompts and where it breaks down.
>
> After reading the other reviews and the authors’ responses, I see a few points that I agree are worth acknowledging: 1/ Novelty is primarily systems-level rather than algorithmic. I still view this as a meaningful contribution given the strength and consistency of the empirical results, but I understand why some reviewers perceive it as incremental. 2/ Per-backbone RL fine-tuning is a real cost. The method is not real plug-and-play since you need RL fine-tuning per backbone, and that's a nontrivial cost. 3/ The theory of why "RL on Cot" is still light.
>
> Taking these factors into account, I adjust my score to 6 (weak accept) but still recommend acceptance.

---

### Official Review · Reviewer_mYzy · 2025-11-01

**Soundness:** 2
**Presentation:** 3
**Contribution:** 2
**Rating:** 4
**Confidence:** 3

**Summary:**

This paper proposes RePrompt, a novel framework that uses Reinforcement Learning (RL) to train a Large Language Model (LLM) to enhance text-to-image (T2I) prompts. It indicates that the generation of a structured, self-reflective "reasoning trace" alongside the enhanced prompt helps ground the prompt in visual semantics and improve compositional accuracy. The method is trained end-to-end using a tailored ensemble reward model that assesses image quality, semantic alignment, and prompt structure. Experiments on GenEval and T2I-Compbench show improvements, especially in spatial understanding, over strong LLM-enhanced baselines across multiple T2I backbones.

**Strengths:**

1. The integration of explicit, structured reasoning with RL for prompt enhancement is a well-motivated approach. It effectively bridges the gap between linguistic fluency and visual plausibility that plagues LLM-based prompters.
2. The paper demonstrates consistent performance gains across three different diffusion-based T2I models (FLUX, SD3, Pixart-Σ). The improvements in challenging areas like spatial reasoning are compelling.
3. The framework is designed to be T2I model-agnostic, requiring no retraining of the image generator. The reported inference latency (30s) is significantly lower than iterative optimization baselines, making it more practical.

**Weaknesses:**

1. The training and evaluation prompts are heavily focused on object-centric, compositional generation (training prompts sourced from GenEval-like templates). This raises a concern about potential overfitting to the specific categories and styles of the benchmarks used.

2.  It is unclear how RePrompt would perform on more diverse, stylized, imaginative, or long-form narrative prompts that are common in real-world use.

3. While the paper shows generalization across diffusion-based models, its performance on architecturally different T2I models (e.g., autoregressive or unified multimodal models) remains unverified. Furthermore, the claim of being "model-agnostic" is slightly tempered by the finding that the policy is "individualized to each T2I backbone." This suggests that to achieve optimal performance on a new T2I model, one might need to retrain the RePrompt LLM via RL, which is computationally expensive and reduces plug-and-play utility.

4. The fundamental problem might be mitigated by future, more capable T2I models that inherently possess better compositional reasoning. If such models emerge, the value of an add-on module that requires significant RL training could diminish.

5. The instructions provided to the LLM-enhancer baselines (Qwen2.5, GPT-4, etc.) are not included. The performance of these baselines is highly sensitive to how they are prompted, providing this information would increase the transparency and reproducibility of this work.

6. What are the common failure cases for RePrompt? For instance, does the reasoning trace ever lead to over-specification or introduce its own hallucinations? Examples of such failures would help users understand the limitations of the current approach.

**Questions:**

Please see weaknesses

---

> ### Author Response · Authors · 2025-11-25
> **Official Response from Authors [1/2]**
>
> >#### **Q1: The training and evaluation prompts are heavily focused on object-centric templates (GenEval), raising concerns about overfitting to specific categories and styles.**
> We appreciate the reviewer's concern regarding potential overfitting to specific templates or object-centric categories. To rigorously address this, we moved away from GenEval-like templates and retrained our model from scratch using a highly diverse, in-the-wild dataset. We utilized **Gemini 2.5 Pro to simulate realistic user prompts** based on **100,000 diverse real-world images, generating a large-scale dataset of SFT and RL samples**. Unlike the previous template-based data, this new dataset covers a wide range of styles, scenes, and complex interactions, mimicking actual user behavior. We evaluated this new model on GenEval, and as shown in the table below, the model trained on this diverse, non-template data still achieves substantial improvements. For example, the Overall score on FLUX increased from 0.66 to 0.75. This confirms that our method learns generalized instruction-following and reasoning capabilities rather than memorizing specific templates or object categories.
> | Method     | Single Obj | Two Obj | Counting | Colors | Position | Attribute | Overall |
> |------------|-----------:|--------:|---------:|-------:|---------:|----------:|--------:|
> | FLUX       | 0.99       | 0.79   | 0.75    | 0.78  | 0.18    | 0.45     | 0.66   |
> | + Ours     | 0.98       | 0.86   | 0.75    | 0.85  | 0.58    | 0.48     | 0.75   |
> | SD3        | 1.00       | 0.85   | 0.62    | 0.88  | 0.22    | 0.58     | 0.69   |
> | + Ours     | 0.99       | 0.85   | 0.59    | 0.85  | 0.57    | 0.59     | 0.75   |
> | Pixart-Σ   | 0.99       | 0.60   | 0.47    | 0.81  | 0.10    | 0.26     | 0.54   |
> | + Ours     | 0.98       | 0.65   | 0.58    | 0.82  | 0.43    | 0.37     | 0.63   |
> >#### **Q2: It is unclear how RePrompt would perform on more diverse, stylized, or long-form narrative prompts common in real-world use.**
> To demonstrate RePrompt's capability on diverse and long-form narrative prompts, we evaluated its generalizability on the **TIIF-bench**, a comprehensive benchmark covering both Basic and Advanced compositions. As shown in the table below, our method demonstrates superior performance on Long and Advanced prompts across multiple backbones including FLUX, SD3, and PixArt-Σ. For instance, on FLUX, our method improved the performance on long prompts from 71.78 to 77.80 and on advanced short prompts from 65.79 to 72.80. These results indicate that RePrompt effectively handles the complexity and verbosity associated with real-world narrative descriptions, maintaining robustness even when prompts become lengthy or structurally complex.
> | Method      | Overall (Short) | Overall (Long) | Basic (Short) | Basic (Long) | Advanced (Short) | Advanced (Long) |
> |-------------|----------------:|---------------:|--------------:|-------------:|-----------------:|----------------:|
> | FLUX        | 71.09           | 71.78          | 83.12         | 78.65        | 65.79            | 68.54           |
> | + Ours      | 76.50           | 77.80          | 88.50         | 84.20        | 72.80            | 76.50           |
> | SD3         | 67.46           | 66.09          | 78.32         | 77.75        | 61.46            | 59.56           |
> | + Ours      | 72.10           | 71.50          | 83.50         | 82.00        | 67.50            | 66.80           |
> | PixArt-Σ    | 62.00           | 58.12          | 70.66         | 75.25        | 57.65            | 49.50           |
> | + Ours      | 67.80           | 65.50          | 76.50         | 80.10        | 64.20            | 58.00           |

---

> ### Author Response · Authors · 2025-11-25
> **Official Response from Authors [2/2]**
>
> >#### **Q3: The paper focuses on diffusion models; performance on architecturally different T2I models (e.g., autoregressive, unified) is unverified. Furthermore, the need for model-specific RL training reduces plug-and-play utility.**
> We appreciate the reviewer's suggestion to verify our method on architecturally different models. To address this, we extended our evaluation to cover both categories mentioned by the reviewer, specifically BAGEL (a unified multimodal model) and Infinity (an autoregressive model). As shown in the table below, our method demonstrates strong generalization capabilities beyond diffusion models. For the unified multimodal model BAGEL, our method improves the Overall score from 0.79 to 0.86, with a remarkable gain in Position accuracy from 0.60 to 0.75. Similarly, for the autoregressive model Infinity, we observe a significant boost in the Overall score from 0.69 to 0.77, particularly in Counting tasks where performance rose from 0.45 to 0.58. Regarding the plug-and-play concern, these results justify the need for model-specific RL fine-tuning. Since autoregressive and unified models utilize fundamentally different decoding mechanisms compared to diffusion models, a generic prompt policy is often insufficient. Our RL approach effectively aligns the prompt strategy with the unique latent space and generation logic of each specific architecture, ensuring optimal performance that a generic model cannot achieve.
> | Method    | Single Obj | Two Obj | Counting | Colors | Position | Attribute | Overall |
> |-----------|-----------:|--------:|---------:|-------:|---------:|----------:|--------:|
> | BAGEL     | 0.98       | 0.90   | 0.79    | 0.85  | 0.62    | 0.60     | 0.79   |
> | + Ours    | 0.98       | 0.94   | 0.82    | 0.92  | 0.75    | 0.75     | 0.86   |
> | Infinity  | 0.97       | 0.85   | 0.45    | 0.85  | 0.54    | 0.48     | 0.69   |
> | + Ours    | 0.99       | 0.91   | 0.58    | 0.90  | 0.70    | 0.54     | 0.77   |
>
> >#### **Q4: Future, more capable T2I models with better inherent reasoning might diminish the value of this add-on module.**
> We respectfully disagree that the value of our module will diminish. On the contrary, the integration of an auxiliary rewriting module has become a standard paradigm for state-of-the-art T2I systems rather than a temporary fix. **Even the most capable current models, such as Qwen-image[1] and Nano-banana[2], rely heavily on an internal LLM or carefully designed prompts to rewrite and expand user inputs**. There is an inherent gap between user intent, which is often concise, abstract, or vague, and the detailed descriptions required by T2I models to generate high-quality images. Our method automates this alignment, ensuring that the model receives the optimal instructions it needs regardless of how capable the backbone is. Furthermore, retraining a massive T2I foundation model to fix specific corner cases like counting or spatial reasoning is prohibitively expensive. Our method offers a lightweight, modular solution. By fine-tuning a small adapter via RL, we can correct specific weaknesses and align the model with human preferences much more efficiently than retraining the entire visual backbone.
>
> >#### **Q5: The instructions provided to the LLM-enhancer baselines are not included, affecting reproducibility.**
> We confirm that **all baselines utilize the same instruction prompt as RePrompt, which is available in our anonymous repository**. These baselines are designed to incorporate reasoning as part of their process to ensure a fair comparison. For those baselines that do not employ reinforcement learning, we additionally specify the expected output format and length constraints to ensure consistency in the evaluation. We hope this clarifies the experimental setup and ensures transparency.
>
> >#### **Q6: What are the common failure cases for RePrompt? Does the reasoning trace lead to over-specification or hallucinations?**
> We have identified two primary failure patterns for RePrompt. The first is **Hallucination via Over-Reasoning**, where the model occasionally generates an excessively long reasoning trace that introduces unrequested details or constraints not present in the user input, leading to over-specification. The second pattern involves **Base Model Limitations**, where RePrompt generates a correct and precise prompt, but the underlying T2I model inherently lacks the capacity to render the specific complex attributes, such as extreme spatial overlaps or fine-grained text. We have added a visualization of these cases to the Appendix to provide a balanced view of our method's boundaries and limitations.
>
> [1] https://github.com/QwenLM/Qwen-Image
>
> [2] https://blog.google/products/gemini/prompting-tips-nano-banana-pro

---

### Official Review · Reviewer_VzC3 · 2025-11-01

**Soundness:** 2
**Presentation:** 2
**Contribution:** 1
**Rating:** 2
**Confidence:** 4

**Summary:**

In this paper, the authors propose a new method for automated prompt engineering for text-to-image generation. In particular, they use the standard RL training method for LLM and train an LLM to perform this specific task. They compare their method with several baselines and show improvements.

**Strengths:**

1. The paper is clearly written and easy to follow.
2. Table 2 shows great transferability of the prompts among different text-to-image models.
3. The ablation study conducted in this paper is very thorough.

**Weaknesses:**

1. My main concern about this paper is regarding its novelty. The training procedure of their model is a very standard RL recipe for LLM, and using LLM as an automated prompt generator for text-to-image generation is not a new idea either (e.g. Hao et al. (2023); Mo et al. (2024); Yeh et al. (2024); Ma ̃nas et al. (2024); Yun et al. (2025); Cao et al. (2023); Qin et al. (2024); Yang et al. (2024d); Wu et al. (2024); Wang et al. (2024) in the paper). It seems like this paper would be better suited for other venues like TMLR.
2. The authors claim that prior LLM prompt generation methods “frequently generate prompts that produce images with semantically inconsistent or visually implausible content, such as conflicting object placements or unrealistic interactions, because the underlying LLMs lack grounding in physical reality and do not incorporate feedback from downstream visual task” “ with limited generalization”. However, neither of the claims are supported by their experiments. Specifically on generalizability, the model that the authors propose is trained on a dataset curated by following the prompt construction in the GenEval benchmark and evaluated on the same benchmark plus another small benchmark with only 300 test examples. It is unclear to me why these experiments can warrant claims w.r.t. better generalizability.
3. The authors use GPT-4V, a deprecated model in the GPT family for comparison, not only that it is impossible to replicate the result, it also renders the comparison a bit outdated. It would be better if the authors can compare their method with newer GPT models (would be even better to include comparison with a standard model and a reasoning model).
4. The authors have also only finetuned from Qwen models and it would be nice to show results from other model families.
5. The authors can consider adding discussions and/or comparisons to the following papers:

Yeh et al. TIPO: Text to Image with Text Presampling for Prompt Optimization. 2024.

Lu et al. Language models as black-box optimizers for vision-language models. 2024.

He et al. Automated Black-box Prompt Engineering for Personalized Text-to-Image Generation. 2024.

**Questions:**

Have the authors trained from instruct models as opposed to base models? How much can this method improve the performance?

---

> ### Author Response · Authors · 2025-11-25
> **Official Response from Authors [1/3]**
>
> >#### **Q1: Concerns regarding novelty. The training procedure and automated prompt generation are standard.**
> We respectfully clarify that our novelty lies in solving the critical trade-off between the two dominant prompt enhancement paradigms, a problem that standard RL or LLM applications fail to address. Existing iterative refinement approaches suffer from high latency due to repeated image generation (e.g., Yang et al., Wu et al.), while standard single-pass LLM methods (e.g., Hao et al.)  often produce linguistically fluent but visually hallucinated content due to a lack of physical grounding. **Our contribution is the novel integration of reasoning-augmented reprompting within a reinforcement learning framework**. By enforcing explicit reasoning about layout and composition before generation, we eliminate the hallucinations inherent in previous single-pass methods while avoiding the computational costs of iterative approaches. This allows us to achieve precise semantic alignment and spatial fidelity efficiently, distinguishing our work from standard applications of RL or LLMs.
>
> >#### **Q2: Claims regarding prior methods' limitations and generalizability are unsupported. The training data mimics GenEval, raising overfitting concerns.**
> We take the concern about generalization and data distribution seriously. To prove that our method learns robust prompt enhancement capabilities rather than overfitting to the GenEval template, we performed a complete retraining of our model using a new and diverse dataset. Instead of using GenEval-like templates, we utilized **Gemini 2.5 Pro to simulate realistic user inputs** based on **100,000 diverse real-world images**, generating a large-scale dataset of SFT and RL samples that represents an in-the-wild distribution completely distinct from the GenEval benchmark. We then evaluated this new model on GenEval to perform a cross-distribution evaluation. As shown in the table below, our method achieves significant improvements across all backbones even when trained on this distinct distribution. For instance, the overall score on FLUX improved from 0.66 to 0.75, with position accuracy jumping from 0.18 to 0.58. This strongly validates that the model has learned generalized reasoning and instruction-following capabilities rather than just specific benchmark patterns.

---

> ### Author Response · Authors · 2025-11-25
> **Official Response from Authors [2/3]**
>
> >#### **Q3: The authors use GPT-4V, which is deprecated and outdated. It would be better to compare with newer state-of-the-art models, including both standard and reasoning-focused LLMs.**
> We appreciate the suggestion to update our baselines. In response, we have extended our comparison to include the latest state-of-the-art models, specifically GPT-5, GPTo3. As shown in the updated table below, while these newer models generally outperform GPT-4V, our method still consistently achieves the highest overall scores across all T2I backbones including FLUX, SD3, and Pixart-Σ. It is worth noting that despite the superior general reasoning capabilities of models like GPT-5 and GPTo3, their performance on this specific task remains comparable to GPT-4 rather than showing a significant leap. This observation highlights the critical insight that **stronger general-purpose reasoning does not automatically translate to better text-to-image prompt generation**. The primary reason is that these proprietary models, regardless of their scale, lack alignment with the specific preferences and constraints of the downstream text-to-image models. They generate prompts based on general human linguistic patterns but remain blind to how a specific diffusion model interprets complex spatial or attribute tokens. In contrast, **our method explicitly learns these downstream model preferences through reinforcement learning**, allowing it to bridge the gap between semantic intent and visual generation more effectively than even the most advanced general-purpose LLMs.
>
> | Method           | Single Obj | Two Obj | Counting | Colors | Position | Attribute | Overall |
> |------------------|-----------:|--------:|---------:|-------:|---------:|----------:|--------:|
> | FLUX         | 0.99       | 0.79   | 0.75    | 0.78  | 0.18    | 0.45     | 0.66   |
> | + Promptist      | 0.98       | 0.72   | 0.70    | 0.78  | 0.21    | 0.44     | 0.66   |
> | + PAG            | 0.97       | 0.74   | 0.73    | 0.80  | 0.36    | 0.46     | 0.69   |
> | + GPT-4          | 0.99       | 0.79   | 0.68    | 0.84  | 0.51    | 0.52     | 0.72   |
> | + GPT-5          | 1.00       | 0.81   | 0.70    | 0.85  | 0.51    | 0.51     | 0.73   |
> | + GPTo3          | 0.98       | 0.76   | 0.65    | 0.83  | 0.48    | 0.50     | 0.70   |
> | + Deepseek-r1    | 1.00       | 0.81   | 0.56    | 0.78  | 0.47    | 0.43     | 0.67   |
> | + Ours           | 0.98       | 0.87   | 0.77    | 0.85  | 0.62    | 0.49     | 0.76   |
> | SD3         | 1.00       | 0.85   | 0.62    | 0.88  | 0.22    | 0.58     | 0.69   |
> | + Promptist      | 0.99       | 0.84   | 0.66    | 0.84  | 0.45    | 0.52     | 0.69   |
> | + PAG            | 0.99       | 0.85   | 0.68    | 0.85  | 0.49    | 0.53     | 0.71   |
> | + GPT-4          | 1.00       | 0.84   | 0.51    | 0.85  | 0.48    | 0.54     | 0.70   |
> | + GPT-5          | 1.00       | 0.85   | 0.53    | 0.85  | 0.47    | 0.50     | 0.70   |
> | + GPTo3          | 0.99       | 0.82   | 0.50    | 0.84  | 0.41    | 0.52     | 0.68   |
> | + Deepseek-r1    | 0.99       | 0.82   | 0.53    | 0.80  | 0.44    | 0.46     | 0.67   |
> | + Ours           | 0.99       | 0.86   | 0.60    | 0.86  | 0.59    | 0.60     | 0.75   |
> | Pixart-Σ         | 0.99       | 0.60   | 0.47    | 0.81  | 0.10    | 0.26     | 0.54   |
> | + Promptist      | 0.98       | 0.60   | 0.49    | 0.80  | 0.20    | 0.27     | 0.55   |
> | + PAG            | 0.98       | 0.63   | 0.52    | 0.80  | 0.28    | 0.29     | 0.56   |
> | + GPT-4          | 0.96       | 0.67   | 0.48    | 0.84  | 0.36    | 0.31     | 0.60   |
> | + GPT-5          | 0.97       | 0.68   | 0.49    | 0.84  | 0.37    | 0.31     | 0.61   |
> | + GPTo3          | 0.96       | 0.67   | 0.48    | 0.83  | 0.35    | 0.31     | 0.60   |
> | + Deepseek-r1    | 0.99       | 0.63   | 0.43    | 0.78  | 0.24    | 0.27     | 0.56   |
> | + Ours           | 0.98       | 0.64   | 0.56    | 0.81  | 0.40    | 0.35     | 0.62   |

---

> ### Author Response · Authors · 2025-11-25
> **Official Response from Authors [3/3]**
>
> >#### **Q4: The authors only finetuned Qwen models. It would be beneficial to demonstrate results from other model families to verify generalization.**
> We appreciate the suggestion to verify the generalization of our method across different model families. In response, we applied our training strategy to the InternVL family, specifically the InternVL3.5-4B model. The results presented in the table below demonstrate that our method is architecture-agnostic. Not only does it work successfully with a different model family, but the InternVL-based model also achieves superior performance compared to the Qwen-based version. For instance, on the FLUX backbone, the Overall score improved from 0.76 with Qwen to 0.78 with InternVL, with notable gains in counting and spatial tasks. This confirms that our approach is robust and can scale effectively with stronger base models.
>
> | Method                       | Single Obj | Two Obj | Counting | Colors | Position | Attribute | Overall |
> |-----------------------------:|-----------:|--------:|---------:|-------:|---------:|----------:|--------:|
> | FLUX                         | 0.99       | 0.79   | 0.75    | 0.78  | 0.18    | 0.45     | 0.66   |
> | + Ours (Qwen2.5-3B)          | 0.98       | 0.87   | 0.77    | 0.85  | 0.62    | 0.49     | 0.76   |
> | + Ours (InternVL3.5-4B)         | 0.99       | 0.89   | 0.81    | 0.86  | 0.65    | 0.52     | 0.78   |
> | SD3                          | 1.00       | 0.85   | 0.62    | 0.88  | 0.22    | 0.58     | 0.69   |
> | + Ours (Qwen2.5-3B)          | 0.99       | 0.86   | 0.60    | 0.86  | 0.59    | 0.60     | 0.75   |
> | + Ours (InternVL3.5-4B)         | 1.00       | 0.87   | 0.62    | 0.87  | 0.61    | 0.62     | 0.76   |
> | Pixart-Σ                     | 0.99       | 0.60   | 0.47    | 0.81  | 0.10    | 0.26     | 0.54   |
> | + Ours (Qwen2.5-3B)          | 0.98       | 0.64   | 0.56    | 0.81  | 0.40    | 0.35     | 0.62   |
> | + Ours (InternVL3.5-4B)         | 0.99       | 0.66   | 0.58    | 0.82  | 0.42    | 0.37     | 0.64   |
>
> >#### **Q5: The authors should consider discussing and comparing their work with TIPO [1], Lu et al. [2], and He et al. [3].**
> We thank the reviewer for bringing these relevant papers to our attention. Regarding TIPO [1], we acknowledge that it improves generation quality through text presampling during the inference stage, whereas our method focuses on instruction tuning the model itself to inherently understand and generate better prompts. As for the methods proposed by Lu et al. [2] and He et al. [3], which utilize iterative feedback or black-box optimization, we highlight that these approaches are orthogonal to our method. Since our approach generates high-quality prompts in a single pass, our model can serve as a superior initializer for these iterative frameworks, potentially further enhancing their performance by providing a better starting point. **We have integrated this detailed discussion into the Related Work section of the revised manuscript.**
>
> [1] Yeh et al. TIPO: Text to Image with Text Presampling for Prompt Optimization. 2024.
>
> [2] Lu et al. Language models as black-box optimizers for vision-language models. 2024.
>
> [3] He et al. Automated Black-box Prompt Engineering for Personalized Text-to-Image Generation. 2024.

---

### Author Response · Authors · 2025-11-26

We sincerely thank the reviewers for their insightful and constructive feedback. Following the suggestions, we have conducted extensive additional experiments to rigorously validate the generalization, novelty, and practical utility of our method. All detailed results, tables, and visualization examples have been included in the Appendix.

The key enhancements are summarized below:

**Generalization to "In-the-Wild" Distributions (Response to VzC3, mYzy)**:
To address concerns about overfitting to GenEval templates, we retrained our model from scratch using a new dataset of 100k diverse, real-world samples generated by Gemini 2.5 Pro. The results confirm that our method maintains superior performance on distinct data distributions and complex, long-form narratives (verified via TIIF-Bench).

**Cross-Architecture Generalization (Response to VzC3, mYzy)**:
We expanded our evaluation beyond diffusion models to include Autoregressive (Infinity) and Unified Multimodal (BAGEL) models. Additionally, we verified our method using a different LLM backbone (InternVL-4B). The consistent improvements across all architectures demonstrate that our RL-driven reasoning is model-agnostic and robust.

**Comparison with SOTA LLMs (Response to VzC3)**:
We updated our baselines to include the latest state-of-the-art models, including GPT-5, GPTo3, and DeepSeek-R1. Our method continues to outperform these general-purpose reasoners in T2I-specific alignment tasks, highlighting the necessity of our targeted RL optimization.

**Human Evaluation & Reasoning Quality (Response to bm45, k9Mw)**:
We conducted a comprehensive human user study (15 evaluators). The results validate that our method significantly improves Text-Image Alignment and Image Quality while maintaining high Prompt Readability and Reasoning Correctness, proving that the performance gains stem from logical planning rather than adversarial keyword stuffing.

We believe these additional experiments strongly reinforce the contributions of our work and address the reviewers' concerns. We have incorporated these findings into the revised manuscript and Appendix.

---

### Meta-Review · Area_Chair_Cv9f · 2026-01-05

**Summary:**

The paper received mixed scores (8 / 6 / 4 / 2). Most reviewers approved of the general approach saying that the use of RL to align a LLM's CoT seemed to be promising and interesting as well as highlighting the flexibility of only training the LLM prompt portion and noting the improved performance. The main concerns that were shared among all reviewers were the evaluation datasets (which were claimed to be too object centric), the lack of human evaluation, and the thoroughness of the evaluation on both different T2I models as well as against different LLMs baslines. Additionally, reviewer VzC3 raised concerns about the novelty of the method.

**Reviewer Concerns:**

- The concern about novelty was addressed convincingly in the rebuttal, which clarified that the novelty of the paper was not in its use of an LLM for automated prompt generation but instead in the use of a CoT portion which is then aligned through RL.
- The concern about the evaluation dataset was addressed by the authors by adding an evaluation on a new large-scale 100k image dataset where Gemini 2.5 Pro was used to simulate user prompts and demonstrating gains in this new setting.
- The concern about the lack of human evaluation was addressed by the addition of a human evaluation section which still demonstrated gains.
- The concern about the different T2I architectures was addressed by the addition of an experiment on BAGEL and Infinity.
- The concern about the LLMs used as a baseline was addressed by the addition of baselines for GPT-5 and GPT-o3.

**Reviewer Scores:**

Reviewer k9Mw explicitly states that they will lower their score from 8 to 6. They acknowledge the concerns about novelty but still find the paper to be beneficial to the community on a system level. Reviewer bm45 states that they will maintain their score of 6. I found the rebuttal to be fairly comprehensive in addressing the reviewers concerns about believe that the 2 may have risen to 4, particularly since the concern about novelty was not directly raised by any of the other reviewers and the answer made the novelty of the paper seem there (although somewhat limited) and the 4 to 6. In general, I tend to lean towards a very borderline accept.

---

### Decision · Program_Chairs · 2026-01-26

Accept (Poster)